# NimA promotes cell adhesion at the blood brain barrier of the *Drosophila* nervous system

Rosy Sakr[1,2,3,4,5,6], Sara Monticelli [ID] [1,2,3,4,6], Smrithi Kizhakkenottiyath Shasthadevan [ID] [1,2,3,4], Claude Delaporte[1,2,3,4], Gege Zhang[1,2,3,4], Tarek Tabiat [ID] [1,2,3,4], Angela Giangrande [ID] [1,2,3,4] & Pierre B Cattenoz [ID] [1,2,3,4]

## Abstract

**Glial cells are crucial for nervous system development and function by clearing debris, protecting neurons and ensuring neuronal survival. In *Drosophila*, glia form the blood–brain barrier, which regulates neural stem cell proliferation and shields the nervous system while maintaining communication with the rest of the organism. To uncover glial-specific roles, we here compare their transcriptome with that of neurons and macrophages. Our study identifies NimA, an uncharacterized member of the Nimrod family, as a glial-specific protein expressed during development. Unlike other family members (i.e. NimC1, Draper and NimC4/Simu) NimA is not involved in phagocytosis. Instead, NimA regulates cell–cell adhesion, crucial for maintaining the tight septate junctions of the larval BBB. Loss of NimA in BBB-forming glia compromises barrier integrity. Moreover, loss of NimA in those glia, or in glia that serve as neural stem cell niche, delays development, reduces brain size, impairs proliferation and reduces the neural stem cell pool. The identification of the glial-specific molecular landscape, including novel molecular players such as NimA, is key for understanding the contribution of glia to the nervous system.**

**Keywords** NimA; Glia; Neurodevelopment; *Drosophila*; Nimrod family
**Subject Categories** Cell Adhesion, Polarity & Cytoskeleton; Neuroscience

## Introduction

*Drosophila* glial cells play essential roles in the development and function of neurons, depending on their position within the nervous system. Typically, perineurial (PG) and subperineurial glia (SPG) of the central nervous system (CNS) form a tight cellular barrier, fulfilling comparable functions to the vertebrate blood–brain barrier (BBB) (Dunton et al, 2021). Within the CNS, the cell bodies of neurons and neural stem cell-like progenitors are insulated and nurtured by cortex glia (CG) (Hartenstein, 2011). Neurons project dendrites and axons in neuropils that are isolated

by ensheathing glia (EG), while axons in the periphery are ensheathed by wrapping glia (WG) (Hartenstein, 2011; Yildirim et al, 2019). Within the neuropils, astrocyte-like glia (ALG) send projections to modulate synaptic activities (Stork et al, 2014). At last, CG, ALG and EG have phagocytic functions at various developmental stages (Hilu-Dadia & Kurant, 2020).

Given the multiple features of glial cells, from neuronal support to immune functions, it is important to define a glial-specific signature. Bulk high-throughput sequencing technologies raise the resolution of cell definition to genomic scale, allowing a global picture of the glial-specific transcriptional landscape. In this study, we determined the molecular signature of *Drosophila* glia upon comparing their transcriptome to that of cells sharing the same ontogeny but have distinct function, i.e., neurons, or have distinct ontogeny but share similar immune and phagocytic functions, i.e., macrophages (also called hemocytes). To extrapolate the most robust and stable features, the comparison was carried out at two stages, at which the three cell types are fully differentiated and functional but embedded in different environments, late embryonic stage (stage 16, E16) and wandering third larval instar (wL3). Embryos represent a closed system in which organogenesis takes place, whereas larvae are constantly exposed to metabolic as well as immune challenges, and at that stage, the growth and development of the adult tissues take place. These data allow us to describe robust glial-specific features that are stable during development. We identify Nimrod A (NimA), which belongs to the family of phagocytic receptors Nimrod (Kurucz et al, 2007; Somogyi et al, 2008). Unlike the other members of the Nimrod family, NimA does not seem to be involved in the regulation of phagocytosis. Instead, we demonstrate that NimA mediates cell aggregation and is required in specific glia subtypes where its loss results in a defective BBB as well as altered neural and animal development.

In sum, we identify the glial-specific transcriptional landscape and unveil NimA as a novel transmembrane protein required to build an efficient cellular barrier.

## Results

### Transcriptomic signature of glia

To define a robust glial-specific signature, we performed a bulk RNA sequencing assay on FACS-sorted glia, neurons and

[1]Institut de Génétique et de Biologie Moléculaire et Cellulaire, Illkirch, France. [2]Centre National de la Recherche Scientifique, UMR7104, Illkirch, France. [3]Institut National de la Santé et de la Recherche Médicale, U1258 Illkirch, France. [4]Université de Strasbourg, Illkirch, France. [5]Present address: Laboratory of Genome Integrity, CCR, NCI, NIH, Bethesda, MD, USA. [6]These authors contributed equally: Rosy Sakr, Sara Monticelli. ✉E-mail: angela@igbmc.fr; cattenoz@igbmc.fr

macrophages at both E16 and wL3 and identified the genes upregulated in glia compared to neurons and macrophages (Figs. 1A, and EV1A,B; Dataset EV1 for genes upregulated in neurons and macrophages, respectively). This comparison highlights known markers that reflect glia origin and function, including the glial cell fate determinant Glial cell missing (Gcm) (Hosoya et al, 1995; Vincent et al, 1996), the pan-glial transcription factor Reversed polarity (Repo)(Halter et al, 1995), the ALG marker Nazgul (Naz) (Ryglewski et al, 2017), the G-protein coupled receptor Moody regulating BBB functions (Bainton et al, 2005) and the WG marker Nervana 2 (Nrv2) (Stork et al, 2008) (Fig. EV1C), validating our approach.

Gene ontology (GO) analyses of the cell-specific signature returned expected terms such as immune response for macrophages and synapse organization for neurons, reflecting the role of these cell types in defense from non-self and in cell circuitry, respectively (Fig. 1B). Interestingly, the GO analyses reveal lower enrichment for glial cells. This likely reflects the fact that specialized glial subtypes perform distinct functions: phagocytosis for ALG, CG and EG, insulation for EG, WG and SPG (Fig. 1B; Dataset EV2). The most significant GO term enriched in glia is cell surface receptors (Fig. 1B), in line with glial cells being in constant interaction with neurons. Amongst the first ten genes identified by this GO (Dataset EV2), the Nimrod A (NimA) transmembrane protein (Fig. 1D), belonging to the family of Nimrod scavenger receptors, displays the strongest enrichment in glia (Fig. 1A').

In sum, our analysis identifies stable cell-specific transcriptomic features that define glia throughout development and highlights NimA as a novel glial-specific marker.

## The glial-specific cell surface protein NimA does not act as a scavenger receptor

In *Drosophila*, the Nimrod family contains 12 members defined by the presence of specific NIM and EGF-like repeats (Kurucz et al, 2007; Somogyi et al, 2008). The characterized Nimrod proteins are involved in cell debris or pathogen recognition to promote phagocytosis. They include the well-studied phagocytic receptors Eater and NimC1 involved in the phagocytosis of pathogens by macrophages (Pearson et al, 1995; Rämet et al, 2001) as well as Draper (Drpr) and Nimrod C4 (NimC4), which act in macrophages and glia to promote the clearance of apoptotic cells and axonal debris (Awasaki et al, 2006; Kurant et al, 2008; Manaka et al, 2004). NimA is the only Nimrod protein specific to glia during development (Figs. 1C and EV1D).

NimA is a Drpr-like protein containing one NIM repeat, two EGF-like repeats and one Emilin (EMI) domain (Figs. 1D and 6) (Callebaut et al, 2003; Kurucz et al, 2007). Its closest mammalian orthologs are Platelet Endothelial Aggregation Receptor 1 (PEAR1) and Multiple EGF-like repeats 10 (MEGF10) that contain multiple EGF repeats and one EMI domain (DIOPT, (Hu et al, 2011)). Beyond the role of PEAR1 in platelet-platelet aggregation (Kauskot et al, 2012; Nanda et al, 2005), both PEAR1 and MEGF10 have been shown to regulate glia-mediated phagocytosis (Iram et al, 2016; Wu et al, 2009), possibly hinting at a functional conservation in phagocytosis between NimA and the two orthologs.

To assess whether NimA is expressed in glial subtypes with known phagocytic functions, we generated a NimA knock-out allele that also acts as a driver (NimA-T2A-Gal4,3xp3mCherry, hereafter NimA KO-Gal4). The NimA coding sequence was replaced by T2A-Gal4 using CRISPR/Cas9 (Fig. 1E). This strategy fully preserves the genomic regulation of NimA expression and produces a faithful NimA Gal4 driver. Of note, NimA KO-Gal4 animals are viable and display curved wings in adults (Fig. EV2A), similar to the phenotype of known NimA null alleles produced by point mutations such as NimA$^{SF7}$ and NimA$^{j-676}$ (Woodruff and Ashburner, 1979).

NimA KO-Gal4 crossed with UAS-fluorescent reporters revealed NimA expression in Repo-positive glia in the late larval (wL3) CNS but not in the peripheral nervous system (Figs. 1F–G' and EV2B–G"). Specifically, NimA is expressed in the BBB glia, in the EG enwrapping the brain and VNC neuropils, as well as in the CG encasing neural stem cells (Figs. 1F–G' and EV2B–D'''). The expression profile of NimA KO-Gal4 is concordant with fluorescent in situ hybridization data obtained using a NimA probe (Fig. EV2H–I''').

Both EG and CG contribute to the phagocytic clearance of apoptotic cells, a process essential for normal CNS development and homeostasis (Hilu-Dadia and Kurant, 2020). A well-characterized example is the removal of the peptidergic Corazonin neurons (vCrz) at early pupal stages. These neurons, arranged in eight bilateral pairs along the larval VNC, undergo apoptosis and are cleared within ~6 h after puparium formation (hAPF) (Choi et al, 2006; Perron et al, 2023; Tasdemir-Yilmaz and Freeman, 2014). Their clearance is mediated by the surrounding glia that express NimA (Fig. 2G,H). Of note, vCrz neuron clearance occurs with similar efficiency in NimA KO-Gal4 mutants (NimA KO-Gal4; UAS-mCD8-GFP) and control animals (UAS-mCD8-GFP or NimA KO-Gal4/+; UAS-mCD8-GFP) at 4 hAPF (Fig. 2A–F,I), indicating that NimA is not required for this phagocytic process. To further assess the phagocytic role of NimA in EG and CG, we used a conditional approach in which a NimA RNAi line (NimA RNAi 104204) was expressed in one or the other glial subtype. We then employed an independent assay based on immunolabeling of cleaved Dcp-1, which marks uncleared apoptotic cells in the third-instar larval CNS (Song, McCall et al, 1997; McLaughlin, Perry-Richardson et al, 2019). Quantification of Dcp-1 signal in wL3 CNS following NimA downregulation in either CG or EG did not reveal significant differences compared with controls (Fig. 2J).

Available single-cell RNA sequencing (scRNAseq) data show that NimA is also detected in EG and CG in the adult brain (Fig. 2K,K') (Davie et al, 2018). We therefore examined whether NimA is required for glial phagocytosis in the adult brain using the clearance of Pigment-dispersing factor–tri (PDF-tri) neurons as a model. PDF-tri neurons appear at mid-pupal stage (Fig. 2L) and undergo apoptosis within three days following hatching of the adult fly (Helfrich-Forster, 1997; Vita et al, 2021). PDF-tri neuron removal requires the scavenger receptor Drpr in the EG and CG wrapping the area (Vita et al, 2021). NimA KO-Gal4 crossed with UAS-fluorescent reporters revealed NimA expression in glia surrounding neuropils (Fig. EV3A–C''') as well as around the PDF-tri neurons in brains from young adults (Fig. 2M–R). However, in 3-day-old adults, the PDF-tri neurons are cleared

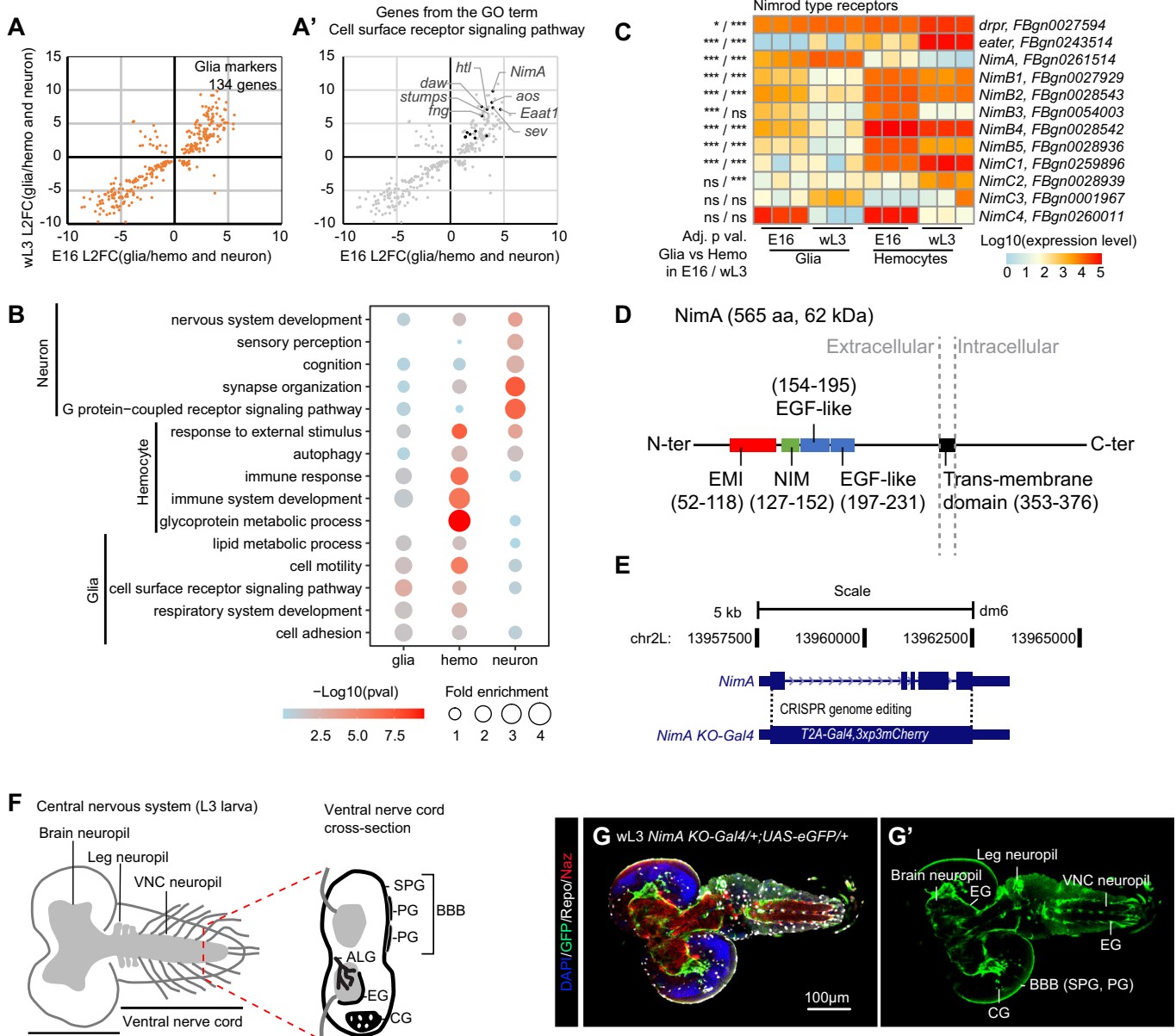

**Figure 1. NimA is a glia-specific protein.**

(A, A') Comparison of the glia transcriptome to the neurons' and macrophages' transcriptomes from stage 16 embryos (E16) and wandering third-instar larvae (wL3). The scatter plot represents the Log2 fold change (L2FC) of differentially expressed genes enriched in glia compared to neurons and macrophages. Only genes presenting significant fold change in both E16 and wL3 were plotted; the x-axes represent the L2FC in E16 and the y-axes the L2FC in wL3. The genes annotated with the GO term "Cell surface receptor signaling pathway" are highlighted in black in (A'). The data were available in Dataset EV1. (B) GO term enrichment analyses on the markers specific to neurons, to macrophages and to glia. The five most significant GO terms are represented here. The size of the dots represents the enrichment levels of the GO term, and the color gradient represents the $p$ value ($-\log10(pval)$, from light blue to red for low to high significance). The data were available in Dataset EV2 and were generated with PANGEA, which calculates enrichment $p$ values via the hypergeometric test and Bonferroni, Benjamini–Hochberg, and Benjamini–Yekutieli procedures. (C) Heatmap representing the expression levels of the Nimrod transmembrane receptors in macrophages and glia. The color gradient is representative of the expression levels from blue to red, low to high expression. Note that most Nimrod receptors are enriched in macrophages, only a few are enriched in glia (i.e., NimA, NimC4). The adjusted $p$ values mentioned in the left column were estimated using the Wald test (using DESeq2) and are issued from the comparison of the expression levels in glia and macrophages at E16 or wL3 (p E16 / p wL3), values in Dataset EV1. (D) Structure of the glial-specific receptor NimA. (E) Schematic representation of the *NimA-T2A-Gal4* allele (*NimA KO-Gal4*). The coding sequence of *NimA* was replaced by *T2A-Gal4*, producing a null allele that also serves as a *Gal4* reporter. (F) Schematic representation of the larval CNS. The perineurial (PG) and subperineurial glia (SPG), composing the blood–brain barrier (BBB), astrocyte-like glia (ALG), ensheathing glia (EG), and cortex glia (CG) are shown on the schematic. (G, G') Immunolabelling of a wandering L3 (wL3) *NimA KO-Gal4/+;UAS-eGFP/+* CNS with the ALG marker Nazgul (in red), the pan-glial marker Repo (in white) and GFP labeling *NimA* cells (in green). Nuclei are labeled with DAPI (in blue). A single section of the larval CNS with all channels merged (G), or the GFP channel (G') are shown. Source data are available online for this figure.

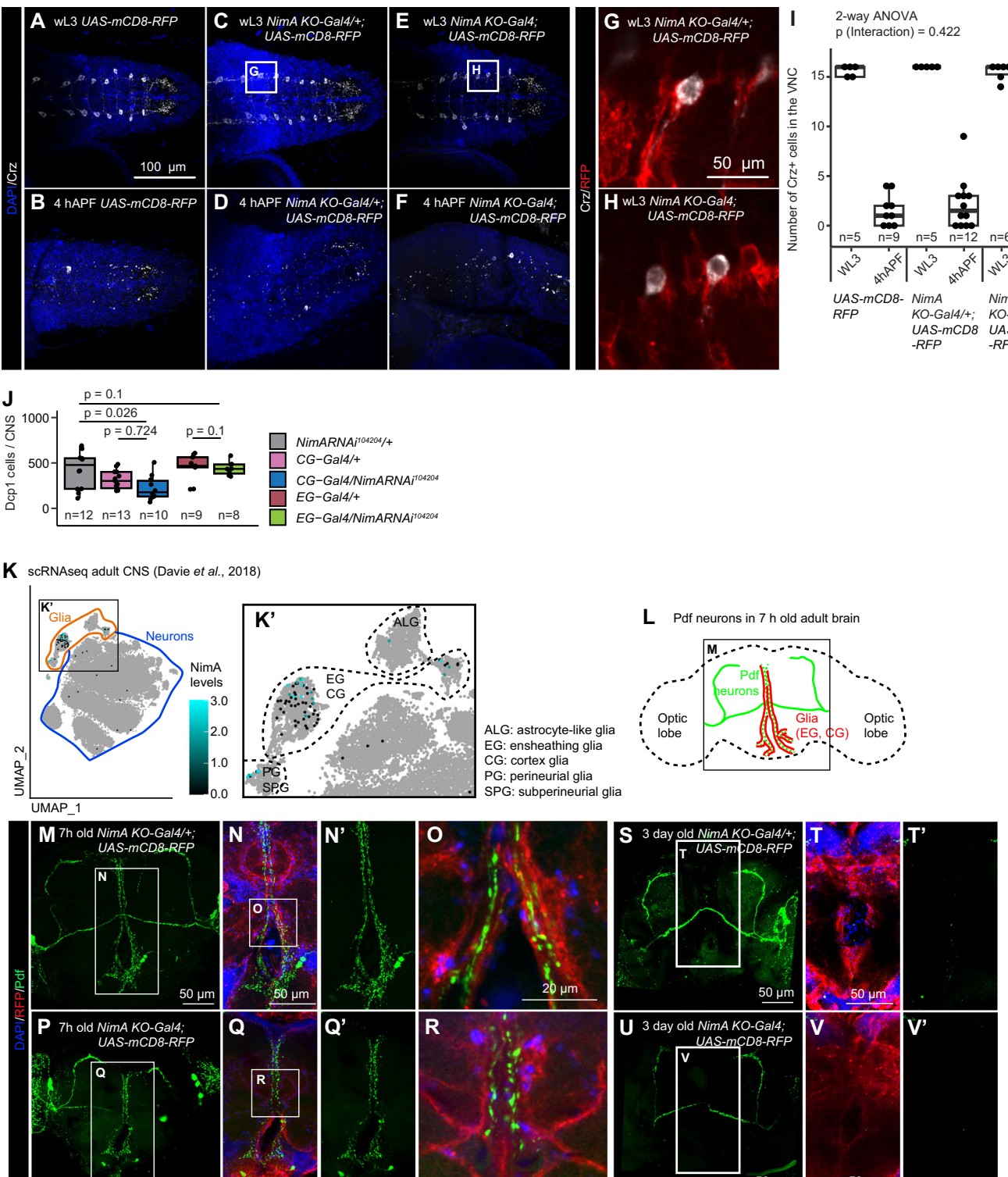

© The Author(s)

in controls as well as in *NimA KO-Gal4* mutants (Figs. 2S–V' and EV3D–J), indicating that NimA has no impact on the phagocytosis of the PDF-tri neuron debris in the adult brain.

In sum, NimA is expressed in multiple glial subtypes (BBB glia, EG, and CG), yet unlike other members of the Nimrod family, it does not appear to be required for glial phagocytosis.

## NimA is expressed in the glia, forming the blood–brain barrier, and is required for its integrity

In addition to their roles in phagocytosis, the scavenger receptors Eater and NimC1 promote cell adhesion and spreading in macrophages (Melcarne et al, 2019b). Given the expression of

◄   **Figure 2.   NimA does not act as a scavenger receptor in glia.**

(A–F) Immunolabelling of control (*UAS-mCD8RFP* or *NimA KO-Gal4/+;UAS-mCD8RFP*) or *NimA KO-Gal4* homozygous (*NimA-T2A-Gal4;UAS-mCD8RFP*) VNC from wandering L3 larvae (wL3, A, C, E, G, H) or pupae 4 h after puparium formation (4 hAPF, B, D, F) with anti-Crz (in gray) that labels the Corazonin neurons undergoing apoptosis in the early pupae. Nuclei are labeled with DAPI (in blue). Full stacks are shown. (G, H) Single sections of the insets indicated in (C, E) highlight the ensheathing glia expressing *NimA KO-Gal4 > mCD8-RFP* (in red) surrounding the Crz neurons (in gray). (I) Quantification of the Crz neurons. Box plots show the median (center line), interquartile range (box), and minimum and maximum values within 1.5 × interquartile range (whiskers). Statistics: two-way ANOVA. n ≥ 5. (J) Quantification of uncleared apoptotic cells in the wandering L3 CNS upon knockdown of NimA (*NimA RNAi104204*) in cortex glia (CG-Gal4: *VGlut2-gal4*) or in ensheathing glia (EG-Gal4: *GMR83E12-Gal4*). Apoptotic debris were identified by immunolabeling with the apoptotic marker Dcp-1 and quantified in 3D using IMARIS software. Box plots show the median (center line), interquartile range (box), and minimum and maximum values within 1.5 × interquartile range (whiskers). Statistics: two-way ANOVA and Tukey HSD post hoc test. n ≥ 8. (K, K′) UMAP representing *NimA* expression profile in scRNAseq data from adult CNS (data and annotation from (Davie et al, 2018)). *NimA* expression levels are represented with a color gradient from black to blue. Higher magnification around the glia cluster is shown in (K′) with annotation for the perineurial (PG) and subperineurial glia (SPG), the astrocyte-like glia (ALG), the ensheathing glia (EG) and the cortex glia (CG). (L) Schematic representation of the PDF-tri neurons surrounded by glia in the adult CNS. (M–V′) Immunolabelling of *NimA* heterozygous (*NimA KO-Gal4/+;UAS-mCD8RFP*) or *NimA KO-Gal4* homozygous (*NimA KO-Gal4;UAS-mCD8RFP*) adult brains, 7 h old (M–R) or 3-day-old (S–V′) with anti-RFP (in red) and anti-Pdf (in green) that labels the PDF-tri neurons undergoing apoptosis within 3 days after eclosion. (M, P, S, U) show full stacks of the brains, (N, Q, T, V) are substacks at higher magnification of the region covering the PDF-tri neurons, the Pdf channels alone are shown in (N′, Q′, T′, V′). Single sections at higher magnifications are shown in (O, R), highlighting the close proximity of *NimA* glia around the PDF-tri neurons. Source data are available online for this figure.

NimA in glia-forming boundaries (EG and BBB glia), we hypothesized that, similarly to NimC1 and Eater, NimA plays a role in the adhesion between glial cells.

In late larvae (wL3), alteration of the EG functions can be visualized by the presence of gaps at the levels of the cellular barrier formed around the VNC neuropils (Pogodalla et al, 2021). No gap nor wrapping defects were detected in the EG surrounding the neuropils of the VNC in *NimA KO-Gal4* homozygous larvae (Fig. 3A–B').

We then monitored the status of the larval BBB, which is formed by PG and SPG, with barrier integrity maintained by septate junctions between SPG cells. When these junctions are compromised, the BBB loses its insulating function (Stork et al, 2008). In situ hybridization assays detected specific *NimA* expression in SPG but not in PG (Fig. 3C–D'''), arguing for a role of NimA in BBB formation and/or maintenance. We estimated BBB permeability by incubating the CNS in 10 kDa dextran coupled with a fluorescent dye that cannot cross the BBB in normal conditions (Winkler et al, 2021). A significant increase in dextran intensity within the CNS of *NimA KO-Gal4* mutants was observed compared to *NimA KO-Gal4* heterozygous and wild-type animals (Fig. 3E–G).

A similar result was obtained upon downregulating *NimA* levels with two independent RNAi lines (*NimA RNAi 104204* and *NimA RNAi 105009* Fig. EV4A) in SPG but not in PG nor CG (Fig. 3H), indicating a specific function of NimA in the BBB glia that form septate junctions. To assess the integrity of the latter, we used the Neurexin-GFP (NrxGFP) transgenic line expressing a fusion protein between GFP and Neurexin IV, which localizes to septate junctions (Baumgartner et al, 1996; Laval et al, 2008) (Fig. 3I–M). *NimA KO-Gal4* mutant CNS displays thinner septate junctions with multiple gaps compared to heterozygous and wild-type CNS (Fig. 3L,M).

These data show that NimA is expressed and required in SPG to build and/or maintain integral septate junctions and preserve the impermeability of the BBB.

## NimA is required for normal neural development

The integrity of the BBB is essential for CNS development, and its alteration is associated with a dysregulation of neural stem cell proliferation in the larva (Speder and Brand, 2014). Accordingly,

the *NimA KO-Gal4* mutants show reduced CNS size at wL3 (Fig. 4A–C), accompanied by a reduced neural stem cell pool and decreased cell proliferation (Fig. 4D,E). Consistent with the observed CNS defects, *NimA KO-Gal4* mutants show developmental delay as most of the animals enter the pupariation stage 1 day later than control animals (Fig. 4F), and a slight increase in larval lethality (Fig. 4J). No further delay or lethality were observed during adult fly eclosion, and the lifespan of *NimA KO-Gal4* mutants is comparable with that of control animals (Fig. EV4B–D).

Given the requirement of NimA in SPG for proper BBB formation, we asked whether the CNS defects were a direct consequence of NimA function in those cells. Indeed, SPG-specific downregulation of NimA is sufficient to reduce CNS volume, proliferation, and neural stem cell number, similar to what we observed in *NimA KO-Gal4* mutants (Fig. 4L–N).

Since NimA is also expressed in CG, which provides a trophic and protective niche for neural stem cells, we analyzed the specific requirement for NimA in those cells as well (Figs. 1G' and EV2C-C'',H–I'') (Bailey et al, 2015; Rujano et al, 2022; Speder and Brand, 2018). Downregulation of *NimA* in CG also leads to reduced CNS size, decreased proliferation, and a smaller neural stem cell pool (Fig. 4L–N). Notably, *NimA* downregulation in either SPG or CG recapitulates the developmental delay seen in *NimA KO-Gal4* larvae (Fig. 4F–H), without affecting viability or adult development (Figs. 4K and EV4E,F,H). In contrast to SPG and CG, silencing *NimA* in EG does not produce detectable CNS or developmental phenotypes (Figs. 4I,K–N and EV4G,H), aside from a very mild delay in adult eclosion (Fig. EV4G).

Overall, these results prove that NimA is required in SPG and CG to support CNS growth.

## NimA promotes cell aggregation and forms multimers

Based on the role of NimA in cells forming barriers, and on the role of its ortholog PEAR1 in stabilizing platelet-platelet interactions (Kauskot et al, 2012; Nanda et al, 2005), we speculated an evolutionarily conserved function of NimA in mediating cell adhesion. We tested this hypothesis in S2 cells transfected with a *NimA* expression vector containing either a Myc (*NimA-Myc-T2A-mCD8GFP*, hereafter *NimA-Myc* gain-of-function, GOF) or an HA (*NimA-HA-T2A-mCD8GFP*, hereafter *NimA-HA GOF*) tag in the

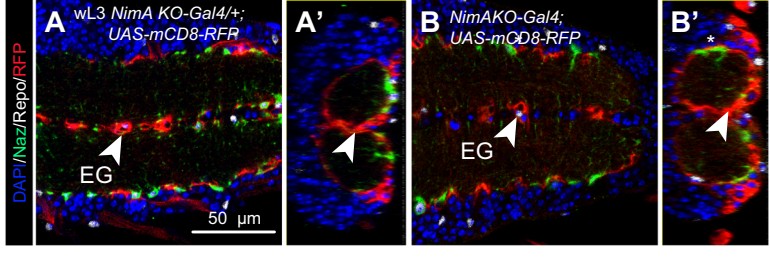

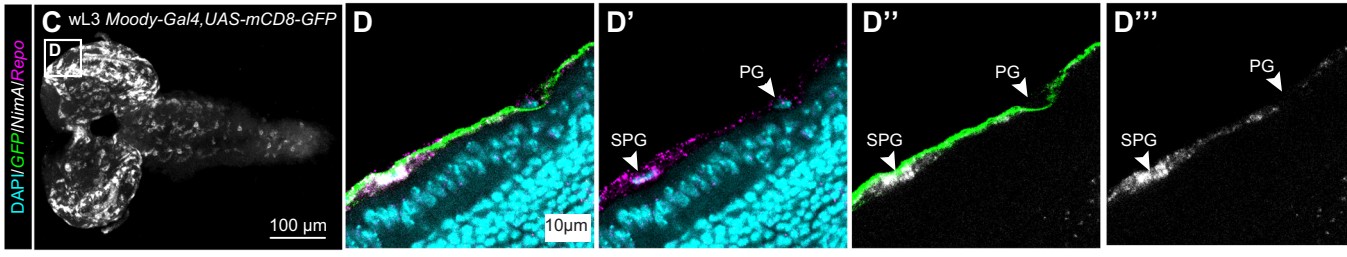

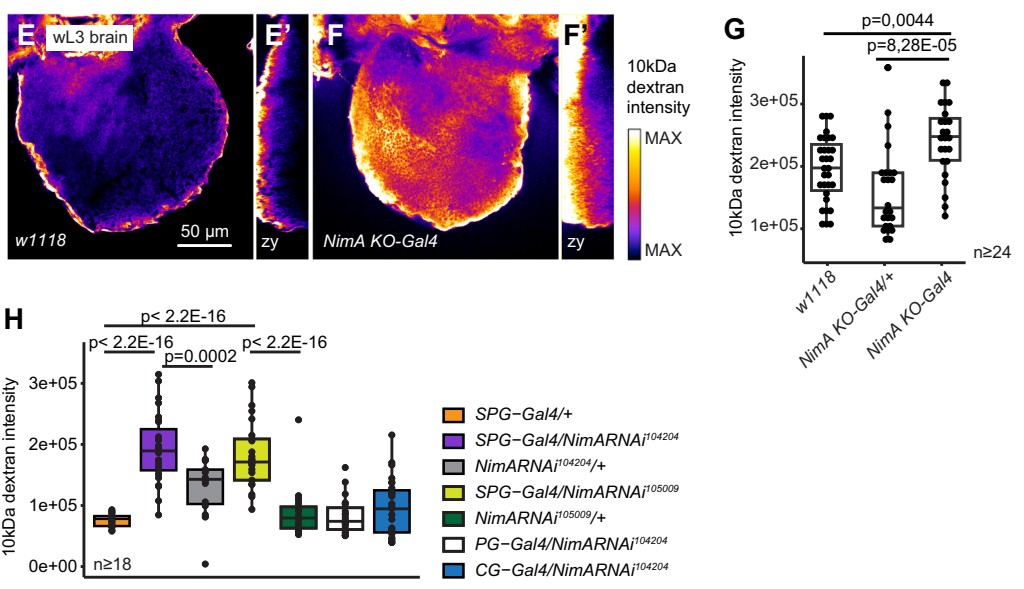

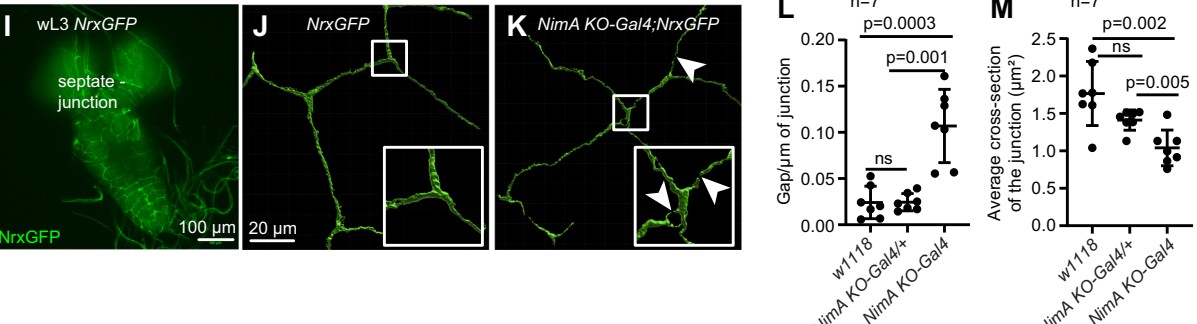

**Figure 3. NimA is required for blood–brain barrier integrity.**

(A, B') Immunolabelling of wandering L3 (wL3) ventral nerve cords from *NimA KO-Gal4/+;UAS-mCD8RFP* (control, **A**) or *NimA KO-Gal4* homozygous (*NimA KO-Gal4;UAS-mCD8RFP*) animals (**B**) with anti-Nazgul (in green), anti-Repo (in white) and anti-RFP (in red). Nuclei are labeled with DAPI (in blue). Single sections with all channels are shown in (**A, B**). Cross-sections at the levels of the arrowheads highlight the ensheathing glia wrapping the neuropils (**A', B'**). (**C, D'''**) Fluorescent in situ hybridization on wL3 CNS from *moody-Gal4,UAS-mCD8GFP* larvae with *NimA* (in white), *Repo* (in magenta) and *GFP* (in green) probes. Nuclei are labeled with DAPI (in cyan). (**C**) Full stack of the *NimA* signal in the whole CNS. (**D–D'''**) Single sections of the inset indicated in (**C**) showing all the channels (**D**), *Repo* and DAPI (**D'**), *NimA* and *GFP* (**D''**), or *NimA* alone (**D'''**). (**E–G**) Penetration assay carried out in wild type (*w1118*, **E, E'**), *NimA KO-Gal4/+* and *NimA KO-Gal4* (**F, F'**) larval CNS with 10 kDa fluorescent dextran. Brains were acquired by confocal microscopy, single sections localized in the middle of the brain are shown in (**E, F**), the Fire color gradient is proportional to the fluorescence intensity from black (low intensity) to white (high intensity). Cross-sections are shown in (**E', F'**). Dextran intensity in the brain is quantified in (**G**). Box plots show the median (center line), interquartile range (box), and minimum and maximum values within 1.5 × interquartile range (whiskers). Statistics: ANOVA with post hoc Student test. $n \geq 24$. (**H**) Penetration assay carried out in wL3 CNS with 10 kDa fluorescent dextran upon downregulating NimA levels in SPG (SPG-Gal4: *Mdr65-Gal4*), PG (PG-Gal4: *shn-Gal4*), or CG (CG-Gal4: *VGlut2-gal4*) by using two independent NimA RNAi lines (*NimA RNAi[105009]* and *NimA RNAi[104204]*) (Kozlov et al, 2020; Mayer et al, 2009). The analysis was performed as in (**E–G**). Box plots show the median (center line), interquartile range (box), and minimum and maximum values within 1.5 × interquartile range (whiskers). $p$ values were estimated by ANOVA with post hoc Student test. $p$ values falling below the R software's numerical precision limit ($p < 2.2E-16$) are indicated as $p < 2.2E-16$. $n \geq 18$. (**I–M**) Analysis of the septate junctions in wL3 brains with Neurexin-GFP (NrxGFP). The fusion protein NrxGFP marks the septate junctions formed by the subperineurial glia (SPG) in the CNS (**I**). (**J, K**) 3D reconstruction of the SPG septate junctions from the brain of *NrxGFP* (**J**) and *NimA KO-Gal4;NrxGFP* (**K**) larvae. Arrowheads indicate gaps in the septate junctions. The number of gaps normalized by the length of the junctions (Gap per µm) is shown in (**L**), and the thickness of the junction (average cross-section) in (**M**). Charts show mean ± standard deviation. Statistics: student tests. $n = 7$. Source data are available online for this figure.

C-terminal intracellular domain (Fig. 5A). NimA localizes at the cytoplasmic membrane, and cytoplasmic projections (Fig. 5D,E), and *NimA GOF* induces cell aggregation, with cells forming large clumps compared to controls (*T2A-mCD8GFP*-transfected S2 cells) (Fig. 5B,C,F).

Since the PEAR1 function requires the EMI extracellular domain (Kauskot et al, 2012), which is the main conserved domain, we assessed whether it is also involved in NimA-mediated cell adhesion. We hence transfected S2 cells with a *NimA* expression vector lacking the EMI domain and containing a C-terminal HA tag (*NimAΔEMI-HA GOF*) (Fig. 5G). Despite the presence of NimA at the cell membrane (Fig. 5H',I'), cell clumps do not increase compared to controls (*T2A-mCD8GFP*-transfected S2 cells) (Fig. 5H–J), indicating that the EMI domain is required for NimA-mediated cell adhesion.

Next, we tested whether the interaction with adjacent cells is accompanied by NimA multimerization, as in the case of PEAR1, where this triggers downstream signaling, stabilizing cell–cell contacts (Kauskot et al, 2012). We co-transfected S2 cells with both *NimA-HA GOF* and *NimA-Myc GOF* vectors or *T2A-mCD8GFP* as a control. NimA-HA and NimA-Myc colocalize at the cell membrane of co-transfected S2 cells (Fig. 5K,L). Immunoprecipitation assays targeting the HA tag, followed by Western blot, show that NimA-HA co-precipitates with NimA-Myc (Fig. 5M). Importantly, different domains account for NimA multimerization and cell adhesion. Indeed, co-precipitates were observed in S2 cells co-transfected with the *NimAΔEMI-HA GOF* and *NimA-Myc* vectors following HA-targeted immunoprecipitation (Fig. 5N), ruling out the involvement of the EMI domain in NimA multimerization.

Overall, these data show that NimA is an evolutionarily conserved transmembrane protein that promotes cell–cell adhesion through the extracellular EMI domain and forms multimers.

## Discussion

In this study, we uncover a robust glial molecular signature and identify NimA as a glia-specific member of the Nimrod family that carries out evolutionarily conserved functions distinct from the canonical scavenger activities associated with other Nimrod receptors. We show that NimA is expressed in multiple central nervous system (CNS) glial subtypes—subperineurial (SPG), cortex (CG), and ensheathing (EG) glia—and contributes to two fundamental biological processes: blood–brain barrier (BBB) integrity and CNS development.

### Expanding the glial molecular landscape

Our comparative transcriptomic analysis of three differentiated and fully functional cell types (glia, neurons, and macrophages) identified 134 genes that define a molecular profile of mature glia preserved throughout development. Beyond known glial markers, we also identified novel ones, including the uncharacterized genes CG1537, CG1545, and CG8837 (Fig. EV1C).

While no predicted functions or orthologs are available for CG1537 and CG1545, CG8837 is suggested to act as a sugar transporter (Ozturk-Colak et al, 2024). Sugar transport in fly glia is fundamental to fuel neuronal function. Perineurial cells of the BBB import sugars from the hemolymph (Volkenhoff et al, 2015). The distribution of metabolic products generated by glial glycolysis, along with direct sugar transport to neurons, maintains CNS homeostasis and functionality (de Tredern et al, 2021). Sugar transport in the CNS is mostly mediated by two solute carrier (SLC) families: SLC5 transporters, which are expressed and required in glia (Yildirim et al, 2022), and SLC2 transporters, so far detected in neurons (Morris et al, 2017; Volkenhoff et al, 2015). CG8837 appears to belong to the SLC2-family of sugar transporters (Ozturk-Colak et al, 2024), consistently with its predicted human orthologs (SLC2A13, SLC2A6, and SLC2A8 (DIOPT (Hu et al, 2011)), all of which are expressed in the mammalian brain, and likely contribute to neuronal function (Byrne et al, 2018; Doege et al, 2000; Ibberson et al, 2002; Ibberson et al, 2000; Uldry et al, 2001). For instance, *Slc2a8–/–* mice exhibit increased proliferation of hippocampal cells and behavioral alterations (Membrez et al, 2006; Schmidt et al, 2008). The myo-inositol transporter SLC2A13 interacts with γ-secretase to induce amyloid-β peptide production (Teranishi et al, 2015), and may modulate neuronal function

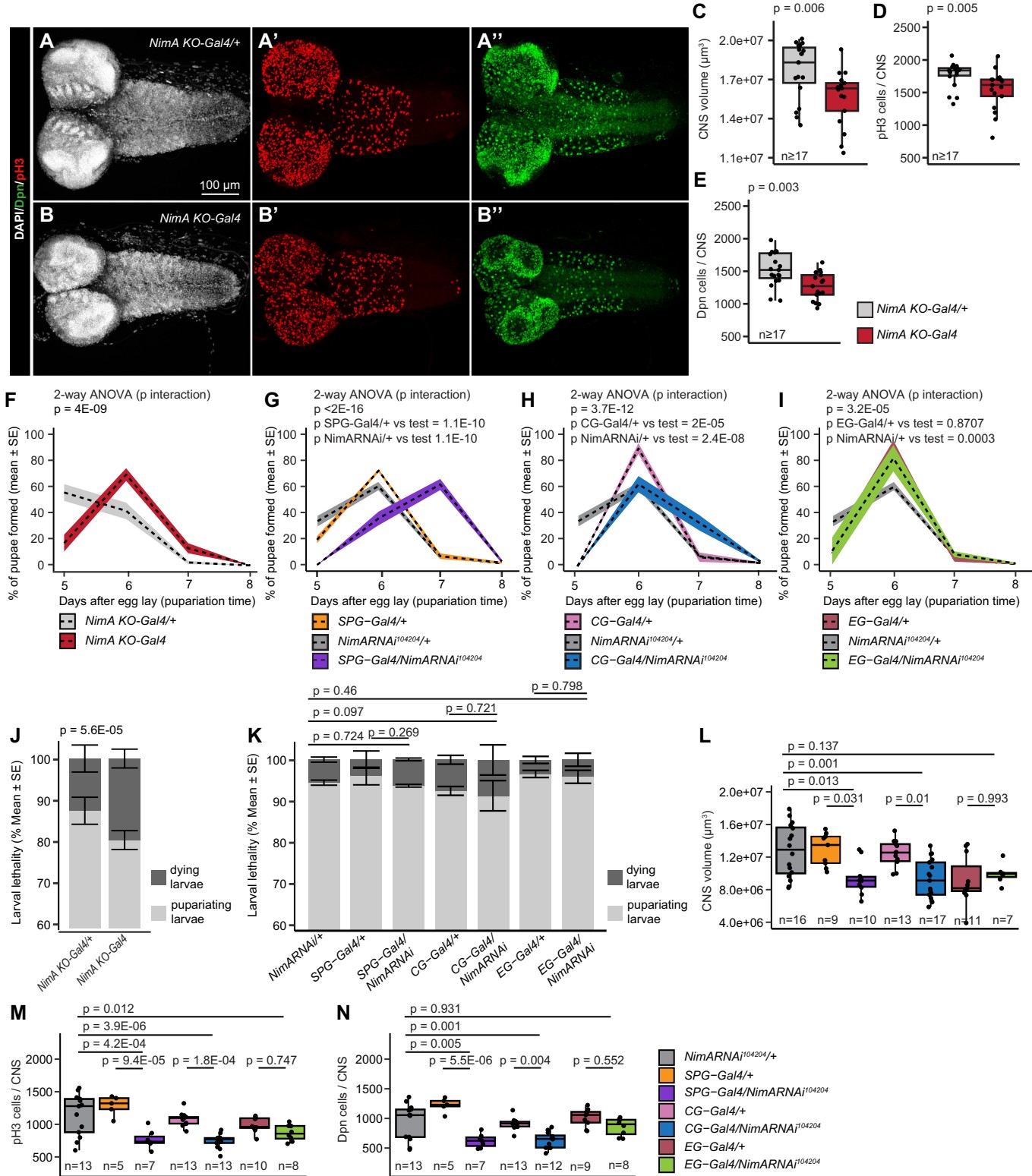

through regulation of brain inositol levels, which are altered in several CNS disorders, including Down syndrome (Berry et al, 1999; Shetty et al, 1995). Future studies will validate CG8837 as an SLC2-family sugar transporter and determine its potential role in glial sugar metabolism.

## NimA's lack of scavenger function: a structural perspective

NimA is expressed in phagocytic-competent glia (EG and CG) and is a member of a well-characterized scavenger receptor family, yet

**Figure 4. NimA contributes to CNS and animal development.**

(A, B) Immunolabelling of wandering L3 (wL3) *NimA KO-Gal4/+* (A) and *NimA KO-Gal4* (B) CNS with antibodies targeting the neural stem cell marker Deadpan (Dpn, in green) and the mitotic marker phospho histone H3 (pH3, in red). Nuclei were labeled with DAPI (in gray). (C–E) Quantification of CNS volume (C), number of mitotic cells marked with anti-pH3 (D), and number of neural stem cells marked with anti-Dpn (E) in *NimA KO-Gal4/+* and *NimA KO-Gal4* wL3 larvae. Box plots show the median (center line), interquartile range (box), and minimum and maximum values within 1.5 × interquartile range (whiskers). Statistics: Mann–Whitney U-test. $n \geq 18$. (F–K) Quantification of pupariation time (F–I) and larval lethality (J, K) in *NimA KO-Gal4* animals vs. controls (*NimA KO-Gal4/+*) (F, J) or upon downregulating NimA levels (*NimA RNAi[104204]*) in subperineurial glia (SPG-Gal4: *Mdr65-Gal4*), cortex glia (CG-Gal4: *VGlut2-gal4*) or in ensheathing glia (EG-Gal4: *GMR83E12-Gal4*) (G–I, K). $n \geq 3$ vials each containing 100 larvae. Statistics: two-way ANOVA (F–I), Chi² test on total dying vs. pupariating larvae (J, K). The same *NimA RNAi[104204]/+* control is represented in (G–I). Charts in (J, K) show mean ± standard error. (L–N) Quantification of CNS volume (L), number of mitotic cells marked with anti-pH3 (M), number of neural stem cell marked with anti-Dpn (N), upon downregulating NimA levels (*NimA RNAi[104204]*) in subperineurial glia (SPG-Gal4: *Mdr65-Gal4*), cortex glia (CG-Gal4: *VGlut2-gal4*), or in ensheathing glia (EG-Gal4: *GMR83E12-Gal4*). Box plots show the median (center line), interquartile range (box), and minimum and maximum values within 1.5 × interquartile range (whiskers). Statistics: two-way ANOVA and Tukey HSD post hoc test. $n \geq 5$. Source data are available online for this figure.

three independent assays performed at larval and adult stages did not reveal any contribution of NimA in promoting phagocytosis in the larval nor adult CNS. One possible explanation for such divergence lies in the structural and evolutionary features of the Nimrod protein family.

Nimrod proteins are characterized by single or multiple copies of the NIM and EGF-like repeats that define their affinity to specific ligands (Fujita et al, 2012; Kurucz et al, 2007; Melcarne et al, 2019a; Melcarne et al, 2019b) (Fig. 6). Drpr contains up to 15 EGF-like repeats (depending on the isoform) and promotes phagocytosis after binding multiple ligands including Pretaporter, phosphatidyl serine exposed by apoptotic cells (Kuraishi et al, 2009; Tung et al, 2013), the chemokine–like protein Orion that coats degenerating neurons, as well as bacteria (Ji et al, 2023; Shiratsuchi et al, 2012). Eater and NimC1, involved in phagocytosis, have extensive extracellular domains with a high number of NIM/EGF-like repeats (Kurucz et al, 2007) (Fig. 6), defining the affinity to distinct bacteria (Kocks et al, 2005). By contrast, NimA displays the shortest extracellular domain with only one NIM and two EGF-like repeats (Kurucz et al, 2007) (Fig. 6). This structural divergence is consistent with the absence of immune or scavenging activity in NimA, which we instead found to be crucial for CNS insulation and development.

## Developmental role of NimA in CNS glia

Among the glial subtypes covering the CNS surface, only SPG, the layer responsible for forming the septate junctions that confer BBB impermeability, expresses NimA. Both in situ hybridization and reporter analyses confirm this restricted expression pattern, and functional assays demonstrate that loss of NimA compromises septate junction architecture and BBB sealing. Moreover, SPG-specific knockdown of NimA is sufficient to phenocopy these defects, demonstrating a cell-autonomous requirement for NimA in these cells.

The physiological consequences of impaired BBB integrity are reflected in the neural phenotypes observed in *NimA* mutants. It has been previously shown that the BBB is essential for relaying nutrient-dependent cues, including insulin signaling, that regulate neural stem cell proliferation (Speder and Brand, 2014). Consistent with this, NimA mutants exhibit a reduced neural stem cell pool and decreased proliferative activity. The basis of neural stem cell loss remains unresolved and may arise from defects in cell fate determination or cell survival (rather than impaired self-renewal), potentially leading to reduced proliferation of progeny cells. These alterations likely contribute to the reduced CNS volume and

developmental delay observed in the mutants. Strikingly, downregulation of NimA in either SPG or CG, which also expresses NimA, is sufficient to recapitulate these defects. While SPG impairment likely disrupts barrier-mediated signaling, the requirement for NimA in CG reveals an independent and previously unrecognized role for NimA in supporting the trophic niche that governs neural stem cell behavior. Notably, BBB-mediated nutritional inputs are essential for CG niche maturation (Speder and Brand, 2018), raising the possibility that NimA in CG and SPG potentially acts within the same pathway. By contrast, despite a clear expression of NimA in EG enwrapping both brain and VNC neuropils, NimA function in EG appears dispensable for CNS and overall animal development, underscoring cell-type–specific requirements. Future studies will elucidate NimA function in EG.

Finally, our analyses indicate that NimA loss does not introduce further developmental delay or lethality past the larval stage, implying that NimA's role is largely confined to developmental stages, with compensatory pathways likely preserving adult homeostasis.

## Potential role of EMI domain and intracellular signaling in NimA-mediated cell adhesion

Our data show that NimA forms homomers and promotes cell–cell contact. The evolutionarily conserved EMI domain appears to mediate cell adhesion, as previously demonstrated for the human ortholog PEAR1 (Kauskot et al, 2012), yet it is not required for NimA homodimerization. This implies that NimA homodimerization alone is not sufficient to drive cell adhesion. NimA-mediated adhesion may require both homodimerization and interactions with additional ligands, potentially engaging the EMI domain.

Such interactions could result either in direct cell–cell adhesion or in the activation of an intracellular signaling cascade, as it happens for other Nimrod proteins. Indeed, following ligand binding, the NimA paralog Drpr is phosphorylated on tyrosine residues in the intracellular domain, which is required for the regulation of phagocytosis (Fujita et al, 2012; Ziegenfuss et al, 2008). Contrarily to the highly structured extracellular domain, the intracellular domain of NimA displays no known protein domain nor conserved feature or structure that could allow inference on its molecular function (Fig. 6). However, several conserved tyrosine residues are predicted to be phosphorylated in the intracellular domain of NimA (Fig. 6, GPS 6.0 prediction score >0.99 (Chen et al, 2023)), raising the possibility that multimerization triggers tyrosine phosphorylation and downstream signaling events responsible for cell aggregation.

In sum, this study identifies a distinctive glial molecular profile, uncovering novel glial markers and a previously undescribed developmental role for NimA, an uncharacterized member of the Nimrod protein family, in BBB integrity and CNS development. Overall, these findings provide new insights into the role and biology of glial cells during development.

# Methods

### Reagents and tools table

| Reagent/resource | Reference or source | Identifier or Catalog Number |
| --- | --- | --- |
| **Fly lines** | | |
| *w1118* | BDSC | 5905 |
| *Oregon-R* | | |
| *elav-nRFP 28.2* | | |
| *repo-nRFP 43.1* | Laneve et al, 2013 | |
| *srp(hemo)Gal4* | Gift from K. Brückner Brückner et al, 2004 | |
| *UAS-RFP* | | |
| *HmlΔ-RFP* | Makhijani et al, 2011 | |
| *srp(hemo)-3xmCherry* | Gift from D. Siekhaus Gyoergy et al, 2018 | |
| *NimA-T2A-Gal4* | This study | |
| *UAS-mCD8RFP* | BDSC | #32218 |
| *UAS-eGFP* | BDSC | #5430 |
| *NrxGFP* | Edenfeld, Volohonsky et al, 2006 | |
| *moody-Gal4,UAS-mCD8-GFP* | Gift from C. Klämbt Stork et al, 2008 | |
| *UAS-NimA RNAi* | VDRC | # 105009 |
| *UAS-NimA RNAi* | VDRC | # 104204 |
| *Mdr65-Gal4* | BDSC | # 50472 |
| *shn-Gal4* | BDSC | # 40436 |
| *VGlut2-gal4* | BDSC | # 39944 |
| *GMR83E12-Gal4* | BDSC | # 40363 |
| Ubi-Gal4 | BDSC | #32551 |
| **Cell lines** | | |
| S2 cells | IGBMC Cell Culture facility | |
| **Recombinant DNA** | | |
| NimA | DGRC | #MIP14095 |
| **Antibodies** | | |
| Chicken anti-GFP | Abcam | #ab13970 |
| Rabbit anti-Nazgul | Gift from B. Altenhein | |
| Mouse anti-Repo | DSHB | #8D12 |
| Rat anti-RFP | Chromotek | #5F8-150 |
| Mouse anti-PDF | DSHB | #C7 |
| Mouse anti-cMyc | IGBMC antibody facility | #Myc 9E10 |

| Reagent/resource | Reference or source | Identifier or Catalog Number |
| --- | --- | --- |
| Rabbit anti-HRP | Cappel | #55974 |
| Rat anti-Dpn | Abcam | #ab195173 |
| Mouse anti-pH3 | Millipore | #05-806 (3H10) |
| Rabbit anti-Dcp-1 | Cell Signaling | #9578S |
| Rabbit anti-HA | Abcam | #ab9110 |
| Rabbit anti-Crz | Gift from J. Veenstra | |
| Phalloidin rhodamine | Cytosqueleton Inc. | # PHDR1 |
| Donkey anti-chicken FITC | Jackson ImmunoResearch | # 703 095 155 |
| Goat anti-mouse Alexa Fluor 647 | Jackson ImmunoResearch | # 115 605 166 |
| Donkey anti-rabbit Cy3 | Jackson ImmunoResearch | # 711-165-152 |
| Goat anti-rat Cy3 | Jackson ImmunoResearch | # 112-165-167 |
| **Quantitative PCR primers** | | |
| *Rp49* F: GACGCTTCAAGGGACAGTATCTG R: AAACGCGGTTCTGCATGAG | Sigma-Aldrich | |
| *Act5C* F: TGCTGCACTCCAAACTTCCA R: GCAGCAACTTCTTCGTCACA | Sigma-Aldrich | |
| *Repo* F: CTCCGCCAAGTAGTTCCTCC R: AGGCAGTAAAGGTGGTTCTCG | Sigma-Aldrich | |
| *Hml* F: CGAGGCAAATCACGATGCTG R: ACGGGCACTTGACGTTGTAT | Sigma-Aldrich | |
| *Drpr* F: GTGGCAGGGTGGGTAGC R: TGATTCATGCCGTAATGTGTGC | Sigma-Aldrich | |
| *Eater* F: ACGATCCATCTAACCGATGTGT R: CGCAGTTATCCTTGCACGTT | Sigma-Aldrich | |
| *NimA* F: AGCCATATGTGGAGCACGTC R: TGACAGCAGAAGCGAACTGT | Sigma-Aldrich | |
| *NimC1* F: TTCGCCATTTTACGGCATGG R: GTCCTGTAGGCAGTCTCATCTT | Sigma-Aldrich | |
| *NimC4* F: CTCGGGCTGAACGAAGCTAT R: CCAAGGGATGAACCTGACCC | Sigma-Aldrich | |
| *NimA* F: CTCCTCCTGCTTGCAATGGT R: TCCTCGCGTATGCAGATGTT | Sigma-Aldrich | |
| **Chemicals, enzymes and other reagents** | | |
| PFA | Electron Microscopy Sciences | 50-980-487 |
| N-phenylthiourea | Sigma-Aldrich | P7629 |

| Reagent/resource | Reference or source | Identifier or Catalog Number |
|---|---|---|
| Collagenase IV | Gibco, Invitrogen) | |
| TRI reagent | Molecular Research Center | TR 118 |
| DNase I recombinant RNase free | Roche | 04716728001 |
| Super-Script IV | Invitrogen | **18090050** |
| SYBR Green I Master | Roche | 04707516001 |
| DAPI | Sigma-Aldrich | Cat# D9542 |
| Vectashield | Vector Laboratories | H-1000-10 |
| Effectene Transfection Reagent | Qiagen | #301427 |
| Proteinase inhibitor cocktail | Roche | # 11697498001 |
| **Software** | | |
| Fiji | RRID:SCR_002285 | |
| IMARIS | RRID:SCR_007370 | |
| R | https://www.r-project.org/ | |
| Adobe Illustrator | RRID:SCR_010279 | |
| HCS Studio software and Colocalisation Bioapplication | Thermo Scientific | |
| **Kits** | | |
| SMARTer low-input RNA kit | Takara | |
| HCR™ RNA-FISH (v3.0) | Molecular Instruments | |

## Fly strains and genetics

Flies were raised on standard medium at 25 °C. Fly strains used are listed in the Reagents and Tools Table. *NimA KO-Gal4* null mutant was generated by CRISPR-mediated mutagenesis and performed by WellGenetics Inc. using modified methods from (Kondo and Ueda, 2013) (Appendix). In brief, the upstream gRNA sequence ACTGCTCCTCCTGCTTGCAA[TGG] and downstream gRNA sequence GTGCCCTTCTAACATATACC[AGG] were cloned into U6 promoter plasmid(s). Cassette *Gal4-3xP3-RFP*, which contains ribosome binding sequence (RBS), Gal4, SV40 polyA terminator and a floxed 3xP3-RFP, and two homology arms were cloned into pUC57-Kan as a donor template for repair. *NimA*-targeting gRNAs and hs-Cas9 were supplied in DNA plasmids, together with a donor plasmid for microinjection into embryos of the control strain *w[1118]*. F1 flies carrying the selection marker of 3xP3-RFP were further validated by genomic PCR and sequencing. CRISPR generates a 4884-bp deletion removing the entire *NimA* CDS that is replaced by the cassette *Gal4-3xP3-RFP*.

## Sample preparation for bulk RNA sequencing

Embryonic neurons, glia and macrophages were isolated from *elav-nRFP* (pan-neural marker), *repo-nRFP* (pan-glial marker) and *srp(hemo)Gal4/+; UAS-RFP/+* (embryonic macrophage marker) E16 embryos upon staged egg laying. After a prelaying of 30 min on apple juice agar plates supplemented with dry yeast, flies were let

lay for 3 h at 25 °C and E16 embryos were collected 11 h and 40 min after egg laying (AEL). Embryos were then washed on a 100-µm mesh and transferred into a cold solution of phosphate-buffered saline (PBS) in a Dounce homogenizer on ice. Embryos were dissociated using the large clearance pestle, then the small clearance pestle, and filtered with a 70-µm filter. wL3 neurons, glia and macrophages were purified from *elav-nRFP*, *repo-nRFP*, and *HmlΔ-RFP/+* (larval macrophage marker) wandering third-instar larvae (L3) collected 108–117 h after a 3-h egg laying at 25 °C. wL3 macrophages were collected upon larva bleeding in cold PBS containing PTU (Sigma-Aldrich P7629) to prevent melanization (Lerner and Fitzpatrick, 1950), and filtered with a 70-µm filter. Neurons and glia were isolated from central nervous systems (CNSs) dissected in cold PBS and transferred in a solution of 0.5 µg of collagenase IV (Gibco, Invitrogen) in 220 µL of PBS. CNS were incubated at 37 °C for 20 min with 500 rpm shaking, dissociated by pipetting with 10-gauge needles and filtered with a 70-µm filter.

Embryonic and larval cells were sorted using FACS Aria II (BD Biosciences) at 4 °C in three independent biological replicates for each genotype. Live cells were first selected based on the forward scatter and side scatter, and only single cells were sorted according to the RFP signal. *Oregon-R* cells were used as a negative control for RFP. Around 100,000 cells were sorted for each replicate. The purity of the sorted populations was assessed by carrying out a post-sort step. The FACS sorter was set up to produce cell pools displaying at least 80% purity on the post-sort analysis.

Cells were collected directly in TRI reagent (MRC) and RNA extraction was carried out following the manufacturer's instructions. Briefly, after 5 min at room temperature (RT) to ensure complete dissociation of nucleoprotein complexes, 0.2 mL of chloroform was added, and samples were centrifuged at 12,000×*g* for 15 min at 4 °C. The upper aqueous phase containing the RNA was collected and transferred to a new tube. 0.5 mL of 2-propanol were added and, after 10 min at RT, samples centrifuged at 12,000×*g* for 10 min to precipitate the RNA. The RNA pellet was then washed with 1 mL of 75% ethanol, then precipitated again at 7500×*g* for 5 min and air dried. About 20 µL of RNase-free water were added to each sample before incubation at 55 °C for 15 min.

Single-end polyA+ RNA-Seq (mRNA-seq) libraries were prepared using the SMARTer (Takara) low-input RNA kit for Illumina sequencing. All samples were sequenced in 50 bp-length Single-Read

## Bulk and single-cell RNA sequencing analysis

The macrophage, glia and neuron bulk RNAseq data produced from E16 embryos and wL3 larvae (wandering third larval instar) were uploaded from EBI database (accession: E-MTAB-8702, (Cattenoz et al, 2020) for macrophages and E-MTAB-14413 (Sakr et al, 2024) for neurons and glia. The raw data were mapped to the *Drosophila* genome August 2014 Dm6 (BDGF release 6 + ISO1 MT/Dm6) using Tophat (Trapnell et al, 2009) and quantified with HT-seq (Anders et al, 2015). For each stage, each tissue was compared to the two others. Log2 fold change and adjusted *p* values were estimated with DESeq2 (Anders and Huber, 2010). Raw read count and Log2 fold change (L2FC) are reported in Dataset EV1. Only genes differentially expressed significantly (adjusted *p* value <0.05) at both stages are represented on (Figs. 1A and EV1A,B). GO term enrichment analysis (Fig. 1B) was done using PANGEA (Hu et al, 2023) using all detected genes as background; the

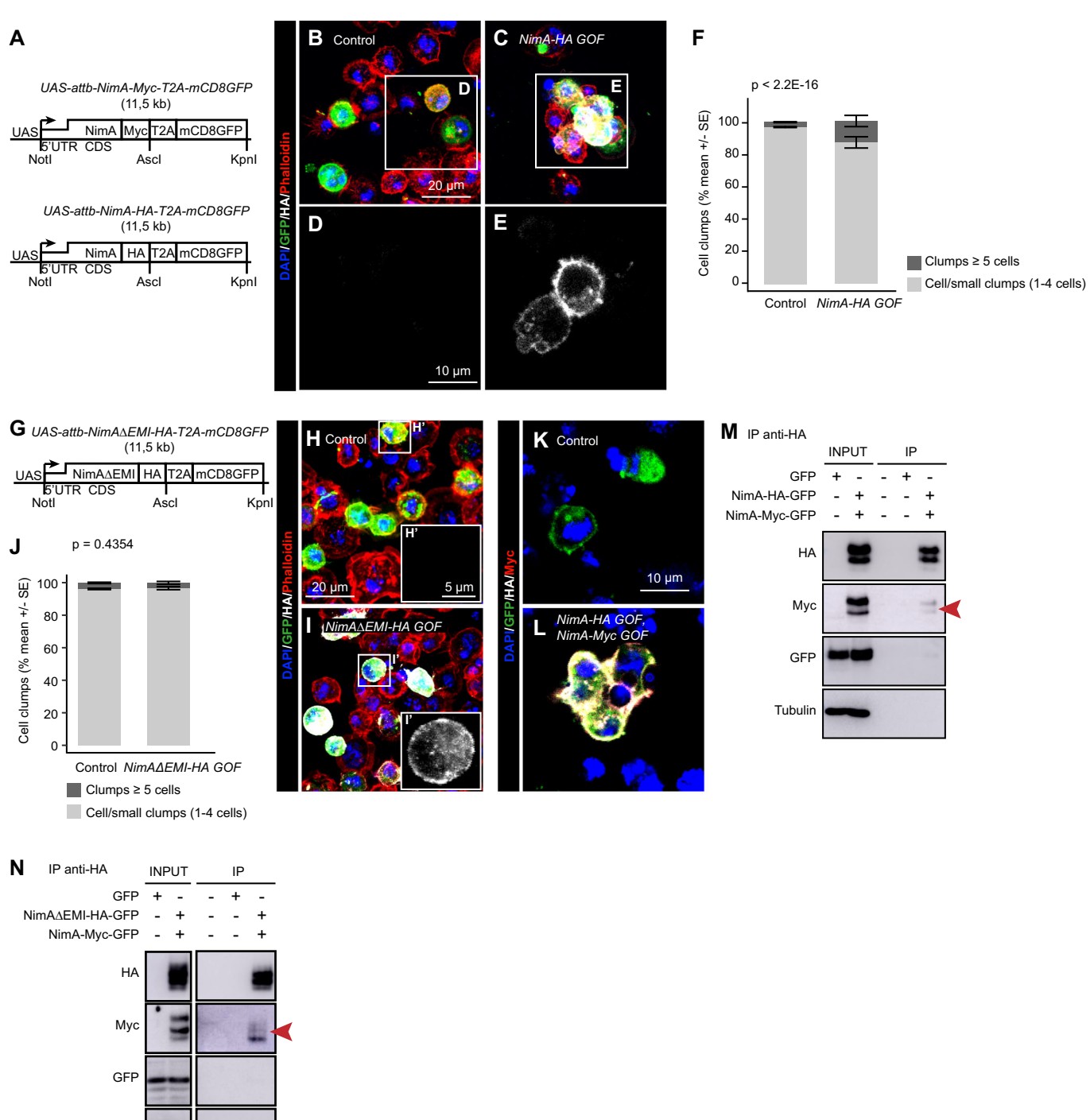

enriched terms are reported in Dataset EV2. The heatmaps (Figs. 1C and EV1C) were plotted in R (version 3.4.0) (R CoreTeam) using the pheatmap (version 0.2) package (https://cran.r-project.org/web/packages/pheatmap/index.html).

The scRNAseq on adult CNS (Fig. 2K,K') were retrieved from (Davie et al, 2018) and the UMAPs were generated with the Seurat package using the FeaturePlot function (Hao et al, 2021; Stuart et al, 2019).

## Quantitative PCR

For the comparison between the expression levels of markers in macrophages and glia (Fig. EV1D), cells were isolated from *srp(hemo)-3xmCherry* and *repo-nRFP* wandering L3 (wL3), respectively, as described in "Sample preparation for bulk RNA sequencing". Upon FACS-sorting in TRI reagent, RNA extraction was carried out as described above. The extracted RNA was treated

◀ **Figure 5. NimA forms homomers and promotes cell aggregation.**

(A) Schematic representation of the expression vectors *NimA-Myc* gain-of-function (*NimA-Myc GOF: UAS-attb-NimA-Myc-T2A-mCD8GFP*) and *NimA-HA GOF* (*UAS-attb-NimA-HA-T2A-mCD8GFP*). (B–E) Immunolabelling of S2 cells transfected with control vector (*UAS-mCD8GFP*, B, D) or *NimA-HA GOF* expression vector (C, E) labeled with anti-GFP (in green) and anti-HA (in white) antibodies and phalloidin (in red). Nuclei are labeled with DAPI (in blue). (B, C) Full stacks of all channels. (D, E) Magnified single sections showing the anti-HA labeling. (F) Quantification of clumps of ≥ 5 cells in control (*UAS-mCD8GFP*) and *NimA-HA GOF*-transfected S2 cells (mean ± standard error). Statistics: Chi² test on total clumps vs. isolated cells/small clumps. *p* value is indicated as *p* < 2.2E-16 because it falls below the R software's numerical precision limit (*p* < 2.2E-16). *n* > 30,000 cells from five independent trials. (G) Schematic representation of the expression vector *NimAΔEMI-HA GOF* (*UAS-attb-NimAΔEMI-HA-T2A-mCD8GFP*). (H, I') Immunolabelling of S2 cells transfected with the control (*UAS-mCD8GFP*, H) or the *NimAΔEMI-HA GOF* vector (I) and labeled with anti-GFP (in green) and anti-HA (in white) antibodies. The F-actin probe phalloidin is in red, and nuclei are labeled with DAPI (in blue). Full stack projections of all channels (H, I) or of the anti-HA signal alone (H', I') are shown. (J) Quantification of clumps of ≥5 cells in control (*UAS-mCD8GFP*) and *NimAΔEMI-HA GOF*-transfected S2 cells (mean ± standard error). Statistics: Chi² test on total clumps vs. isolated cells/small clumps. *n* > 30,000 cells from five independent trials. (K, L) Immunolabelling of S2 cells transfected with the control (*UAS-mCD8GFP*, K) or the *NimA-HA GOF* and *NimA-Myc GOF* vectors (L) and labeled with anti-GFP (in green), anti-HA (in white) and anti-Myc (in red). Nuclei are labeled with DAPI (in blue). Single-section images are shown. (M) Co-immunoprecipitation assay from S2 cells co-transfected with the *NimA-HA GOF* and *NimA-Myc GOF* vectors. The immunoprecipitation (IP) was carried out with anti-HA antibody, and HA, Myc, GFP, and Tubulin levels were detected in the input and in the IP by Western blot. The two first lanes represent the input of the control (*UAS-mCD8GFP* vector), and the *NimA-HA GOF* and *NimA-Myc GOF* co-transfected cells. The three following lanes represent IPs on beads incubated with lysis buffer only, control cells and co-transfected cells, respectively. (N) Co-immunoprecipitation assay from S2 cells co-transfected with the *NimAΔEMI-HA GOF* and *NimA-Myc GOF* vectors. The IP was carried out with anti-HA antibody, and HA, Myc, GFP, and Tubulin levels were detected in the input and in the IP by Western blot. The two first lanes represent the input of the control (*UAS-mCD8GFP*) and the *NimAΔEMI-HA GOF* and *NimA-Myc GOF* co-transfected cells. The three following lanes represent IPs on beads incubated with lysis buffer only, control cells and co-transfected cells, respectively. Source data are available online for this figure.

with DNase I recombinant RNase free (Roche) and the reverse transcription was done using the Super-Script IV (Invitrogen) with random primers. The cycle program used for the reverse transcription is 65 °C for 10 min, 55 °C for 20 min, 80 °C for 10 min. The qPCR was done using SYBR Green I Master (Roche). Actin5C (Act5C) and Ribosomal protein 49 (Rp49) were used to normalize the data.

To assess the expression levels of NimA upon its ubiquitous downregulation via RNAi (Fig. EV4A), RNA was extracted from whole wL3 larvae. RNA extraction, DNase treatment, reverse transcription and qPCR were performed as described above.

The primers are listed in the Reagents and Tools Table. The *p* values and statistical test used are indicated in the figure legends.

## Immunolabeling and image analysis of CNS (larva, adult) and of *Drosophila* S2 cells

wL3 larvae were staged using a 4 h egg lay at 25 °C. L1 larvae were collected 24 h AEL. For each genotype, 100 L1 were transferred to a fresh medium vial and incubated at 25 °C until the wL3 stage. NimA mutant or knockdown larvae were allowed to develop until the same stage as controls. Adults were collected at 7 h or 3 days post eclosion for the phagocytic assay on PDF neurons. wL3 and adult CNSs from the indicated genotypes were dissected in cold PBS 1x and transferred into wells containing 4% PFA in 1x PBS and fixed for 2 h at RT. They were then washed in PTX (0.3% Triton X-100 in 1x PBS) for 1 h and incubated in blocking reagent for 1 h at RT (O/N incubation at 4 °C in blocking reagent for labeling with Dcp-1 and deadpan). CNSs were then incubated with primary antibodies (Reagents and Tools Table) O/N at 4 °C, washed in PTX 3×10 min and incubated with secondary antibodies (Jackson) for 1 h at RT. After washing in PTX and labeling with DAPI, CNSs were mounted in Vectashield.

Transfected *Drosophila* S2 cells were fixed for 10 min with 4% PFA in 1x PBS at RT, washed for 10 min in PTX at RT, incubated for 1 h in blocking reagent at RT and incubated with the indicated primary antibodies diluted in blocking reagent overnight (O/N) at 4 °C. The following day, samples were washed 10 min in PTX at RT, incubated in secondary antibodies diluted 1:400 in blocking reagent for 1 h at RT and incubated in DAPI with or without phalloidin

Rhodamine for 30 min at RT. Samples were finally washed in 1x PBS, mounted in Vectashield.

Images were acquired with a Leica SP8 inverted-based confocal microscope or Leica spinning disk microscope. Image analysis was carried out using Fiji (National Institute of Mental Health, Bethesda, Maryland, USA, (Schindelin et al, 2012)). The surface of the PDF-tri neurons in the adult brain (Fig. EV3J) was estimated with Fiji on the max projection of 15 sections (2-μm thick) abutting the transversal PDF-tri neurons. The CNS volume, as well as the number of pH3/Dpn/Dcp-1 positive cells in the CNS, were estimated with the IMARIS software (RRID:SCR_007370) based on confocal images.

## In situ hybridization

We used an HCR™ RNA-FISH (v3.0) (Molecular Instruments) reagent kit and probes (*NimA, Repo, GFP*). CNSs from *moody-Gal4,UAS-mCD8GFP* wL3 larvae were dissected in cold 1x PBS, fixed in 4% PFA diluted in 1x PBS for 30 min, washed 5 × 10 min in PTX (0.3% Triton X-100 in 1x PBS), then treated according to manufacturer instructions. Briefly, they were incubated in hybridization buffer for 30 min at 37 °C, then in the probe solution (anti-*NimA*-B1, anti-*Repo*-B3, and anti-*GFP*-B2) O/N at 37 °C. The next day, after 4 × 15 min washes in wash buffer at 37 °C and 2 × 5 min washes in SSCT (0.1% Tween 20 in 5x SSC) at RT, samples were incubated 10 min in amplification buffer at RT, then O/N with the amplifiers (B1-647, B3-546, and B2-488) at RT. Finally, samples were washed 2 × 5 min, 2 × 30 min, and 1 × 5min in SSCT at RT, labeled with 1:5000 DAPI diluted in PTX for 30 min at RT and mounted in Vectashield.

## Blood–brain barrier permeability assay

Animals were staged with a 4 h egg lay at 25 °C. L1 were collected 24 h AEL. For each genotype, 100 L1 were transferred to a fresh medium vial and incubated at 25 °C until the wL3 stage. wL3 were dissected in cold PBS 1X and incubated for 30 min in 0.25 mM of 10 kDa dextran Texas red (Invitrogen). The CNSs were washed two times of 10 min in 4% PFA at RT with agitation (50 rpm) to remove excess dextran, followed by incubation in 4% PFA for 1 h and

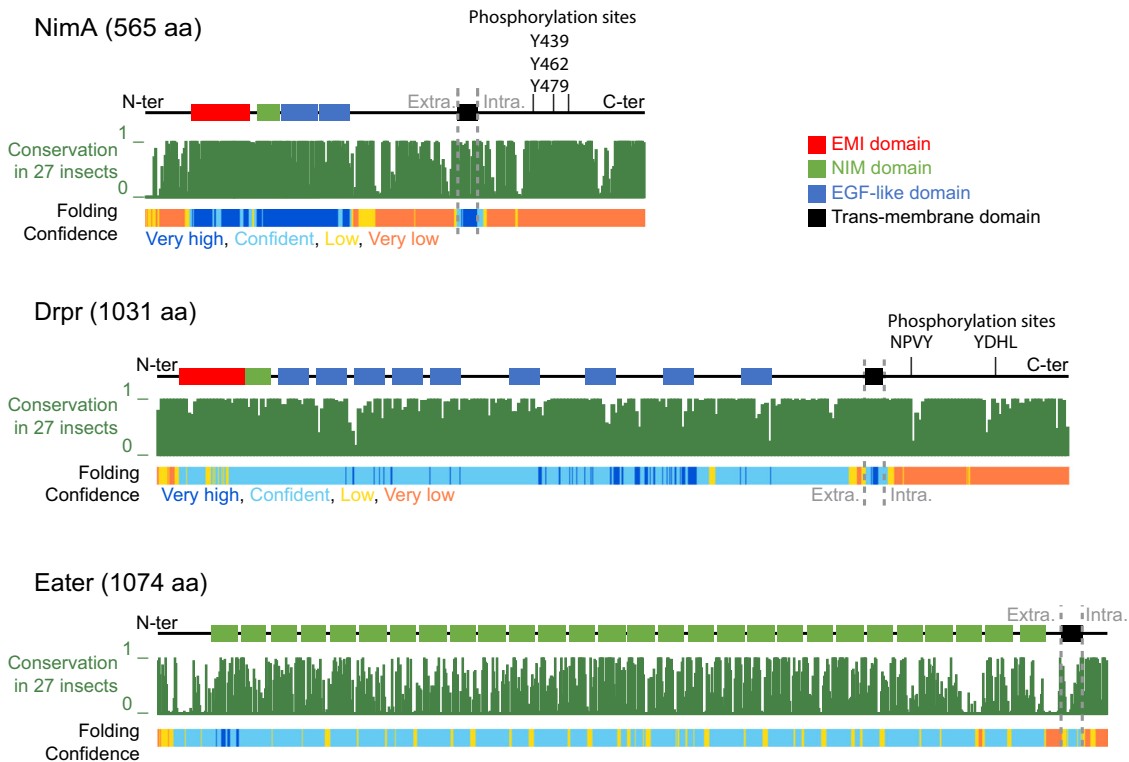

**Figure 6. NimA has a shorter extracellular domain than the phagocytic receptors Drpr and Eater.**

Conservation and structure prediction of Nim, its paralog Drpr and the Nimrod receptor Eater. The conservation histograms (in green) across 27 insect species were drawn from the UCSC genome browser track Multiz Alignment & Conservation on the *Drosophila* genome Dm6. The protein domain annotations and folding prediction were taken from Uniprot and alphaFold protein structure database on the proteins NimA (Q8IP58), Drpr (Q9W0A0) and Eater (Q9VB78). Tyrosine phosphorylation sites are indicated in the intracellular domains of NimA and Drpr. For NimA, the sites were predicted with GPS 6.0 (prediction score >0.99) (Chen et al, 2023).

40 min. The CNS were mounted in Vectashield and acquired using a Leica spinning disk microscope with a 63x objective and 1-μm confocal section. The dextran intensity was measured using Fiji (National Institute of Mental Health, Bethesda, Maryland, USA (Schindelin et al, 2012).

## Septate junction analysis with *NrxGFP* animals

Larvae *NrxIVGFP* and *NimA-T2A-Gal4/+;NrxIVGFP* were used as controls and *NimA-T2A-Gal4;NrxIVGFP* as *NimA KO-Gal4*. Animals were staged with a 4 h egg lay at 25 °C. L1 were collected 24 h AEL. For each genotype, 100 L1 were transferred to a fresh medium vial and incubated at 25 °C until the wL3 stage. The wL3 were dissected in cold PBS 1X carefully without touching the CNSs. Then, CNSs were incubated for 30 min in 4% PFA at RT on the shaker. The CNSs were washed three times for 10 min with 0.3% PTX, incubated with blocking reagent for 1 h at RT on the shaker, incubated overnight at 4 °C with primary antibodies, washed three times for 10 min with 0.3% PTX, incubated for 1 h with secondary antibodies, washed two times for 10 min with PTX, incubated for 40 min with DAPI and then mounted in Vectashield. The following primary antibody was used to label NrxGFP: Chicken anti-GFP (1/1000). Secondary antibody used was: donkey anti-chicken coupled with FITC (1/400). The central area of the brain was imaged using a Leica spinning disk microscope with a 63x oil objective with 0.2-μm

confocal section. The junctions were analysed using Imaris (Version 10.1.1) to reconstruct the BBB junctions. The filament statistics, number of segments, and filament volume were measured on the 3D reconstruction for NrxGFP at the brain surface. Then, the ratios of [(number of segments −1)/filament length] (i.e., gap per μm) and [volume of the filament/filament length] (i.e., average cross-section area) were determined for each brain lobe.

## Developmental delay/lethality and lifespan assay

To score for developmental delay and lethality, animals were let to lay eggs for 4 h at 25 °C on apple juice agar plates supplemented with dry yeast. *w;NimA-T2A-Gal4* (*NimA KO-Gal4*, experimental genotype) or *w;NimA-T2A-Gal4/+* (control genotype, F1 from *NimA KO-Gal4* crossed with *w1118*) L1 larvae were collected 24 h AEL. For each genotype, 100 L1 were transferred to a fresh medium vial and let develop at 25 °C. The number of newly formed pupae and eclosed adults was monitored daily. At least three vials (≥300 larvae) were analyzed per genotype.

To assess the lifespan, 1-day-old animals (*w;NimA-T2A-Gal4* or *w;NimA-T2A-Gal4/+*) were collected and transferred to a fresh medium vial so that each vial contained ten males and ten females. Vials were flipped three times per week for 60 days, and the presence of dead flies was monitored at each flipping. Five vials of 20 adults were analyzed per genotype (n = 100).

## NimA expression vectors

To generate the vector *UAS-attb-NimA-T2A-mCD8GFP*, the NimA coding sequence was retrieved from *pOT2 NimA-RD* (DGRC #MIP14095) using primers containing NotI restriction site for the Forward primer targeting the 5' end of NimA and AscI restriction site on the reverse primer targeting the codon stop of NimA coding sequence. The PCR was carried out using high-fidelity Taq polymerase (Thermo Fisher Scientific). The mCD8GFP coding sequence was retrieved from genomic DNA extracted from the fly stock *QUAS-mCD8GFP* (BDRC #30001) using primers including AscI restriction site and the sequence coding for the self-cleaving peptide T2A on the forward primer and the restriction site KpnI on the reverse primer. The two PCR products were digested with their respective restriction enzymes and inserted in the *UAS-attb* vector (DGRC Stock 1419; https://dgrc.bio.indiana.edu//stock/1419; RRID:DGRC_1419) digested with NotI and KpnI.

For the vector *UAS-attb-NimA-Myc-T2A-mCD8GFP*, the Myc sequence was inserted into *UAS-attb-NimA-T2A-mCD8GFP* by PCR. To generate *UAS-attb-NimA-HA-T2A-mCD8GFP*, the Myc tag was replaced by the HA by PCR. To generate the *UAS-attb-NimAΔEMI-HA-T2A-mCD8GFP*, the EMI domain in the *UAS-attb-NimA-HA-T2A-mCD8GFP* plasmid was removed by PCR.

The *UAS-attb-T2A-mCD8GFP* vector was produced by inserting the *T2A-mCD8GFP* sequence into the *UAS-attb* empty vector.

The sequence of each plasmid was verified by Sanger sequencing.

## Aggregation assays

To estimate the aggregation of the cells, three million *Drosophila* S2 cells were plated per well in a 6-well plate with 1.5 mL of Schneider medium + 10% FCS + 0.5% PS. Transfections were carried out 1 h after plating, using the Effectene Transfection Reagent (Qiagen #301427) according to the manufacturer's instructions. About 0.5 µg of Gal4 expression vector (SKactin-Gal4, RRID:DGRC_1019) was co-transfected with 0.5 µg of vector *UAS-attb-NimA-HA-T2A-mCD8GFP* (*NimA-HA GOF*) or 0.5 µg of vector *UAS-attb-NimAΔEMI-HA-T2A-mCD8GFP* (*NimAΔEMI-HA GOF*), or 0.5 µg of vector *UAS-attb-T2A-mCD8GFP* (control). Transfection was carried out for at least 48 h at 28 °C. Cells were then fixed with 4% PFA in 1x PBS, immunolabelled with anti-GFP and DAPI as described above and scanned using the CellInsight CX7 instrument (Cellomics) at 20x magnification (60 fields per well). The GFP signal was used to determine the boundary of the clumps, and DAPI was used to count the number of nuclei per clump. The signal was estimated using HCS Studio software and Colocalisation Bioapplication.

## Co-immunoprecipitation (IP) and immunoblot analysis

About $3 \times 10^6$ *Drosophila* S2 cells were plated per well in a six-well plate with 1 mL of Schneider medium + 10% FCS + 0.5% PS. Transfections were carried out 1 h after plating, using the Effectene Transfection Reagent (Qiagen #301427) according to the manufacturer's instructions. For the "NimA-HA, NimA-Myc GOF" sample, cells were co-transfected with *SKactin-Gal4, UAS-attb-NimA-HA-T2A-mCD8GFP* and *UAS-attb-NimA-Myc-T2A-mCD8GFP*, 0.3 µg each. For the "NimAΔEMI-HA, NimA-Myc

GOF" sample, cells were co-transfected with *SKactin-Gal4, UAS-attb-NimAΔEMI-HA-T2A-mCD8GFP* and *UAS-attb-NimA-Myc-T2A-mCD8GFP*, 0.3 µg each. Control cells were co-transfected with *SKactin-Gal4* and *UAS-attb-T2A-mCD8GFP*, 0.5 µg each. Transfections were carried out for 72 h at 28 °C.

At least $6 \times 10^6$ transfected cells were used for Co-IP experiments. Cells were washed in 1x PBS and resuspended in 500 µL of lysis buffer (50 mM Tris-HCl, pH 7.5, 150 mM NaCl, 10% glycerol, 1 mM EDTA, 1% Triton, 1x proteinase inhibitor cocktail (Roche). Samples were incubated 10 min on ice, then centrifuged at 10,000×g for 5 min at 4 °C. 10% of the cell lysate was used as Input, complemented with 2X SDS buffer containing 0.2 M DTT and boiled at 95 °C for 5 min. The rest of the cell extract was used for IP with anti-HA agarose beads (Monoclonal anti-HA-agarose antibody produced in mouse, Sigma #A2095). Before use, beads were washed three times in lysis buffer and resuspended in 500 µL of lysis buffer. A "bead control" sample was included in each experiment upon incubating the beads with 950 µL of lysis buffer. IP samples were incubated O/N at 4 °C in agitation, washed three times in lysis buffer, eluted in DTT-free, 2X SDS buffer and boiled at 95 °C for 5 min. Beads were removed by centrifugation, and 0.1 M DTT was added. INPUT and IP samples were then processed by Western blot.

Proteins were separated with a 10% SDS–PAGE gel, transferred onto a nitrocellulose membrane and probed with primary antibodies diluted in 2% milk in 1x PBS. The following antibodies were used: mouse anti-Myc (1:1000), mouse anti-HA (1:1000), mouse anti-Tubulin (1:2000), chicken anti-GFP (1:1000). Signals were detected using anti-mouse and anti-chicken HRP-conjugated antibodies (1:5000) and the chemiluminescent substrate Super-Signal West Pico PLUS (Thermo Fisher Scientific #34580). Chemiluminescence detection was performed Amersham Imager 600.

## Statement on the experimental design

All experiments were conducted using independent biological replicates, with the number of replicates specified in the corresponding figure. No randomization or blinding procedures were applied. Both male and female subjects were included in each experiment unless otherwise stated.

# Data availability

The RNA-seq data for the glia and neurons are available in the European Nucleotide Archive, accession: E-MTAB-14413 (https://www.ebi.ac.uk/ena/browser/view/PRJEB79719). Data Ref: (Sakr et al, 2024).

The source data of this paper are collected in the following database record: biostudies:S-SCDT-10_1038-S44319-026-00728-1.

# Peer review information

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

## Acknowledgements

We thank Lara El Berjawi, Valentine Jungmichel, Nafiseh Pisheh, Luca Sartori, Alexia Pavlidaki, and Maria Dolores De Donno for technical help.

We acknowledge the Imaging Center of the IGBMC, a member of the national infrastructure France-BioImaging supported by the French National Research Agency (ANR-10-INBS-04). The sequencing was performed by the GenomEast platform, a member of the "France Génomique" consortium (ANR-10-INBS-0009). We thank A. Maglott-Roth from the IGBMC screening facility for the macrophage analysis. We thank D. Siekhaus, J. Veenstra and B. Altenhein for providing fly stocks and antibodies. In addition, stocks obtained from the Bloomington Drosophila Stock Center (NIH P40OD018537) and antibodies obtained from the Developmental Studies Hybridoma Bank created by the NICHD of the NIH and maintained at The University of Iowa (Department of Biology, Iowa City, IA 52242) were used in this study. This work was supported by Inserm, CNRS, University of Strasbourg, Ligue Régionale contre le Cancer, Hôpital de Strasbourg, ARC, Indo-French Centre for the Promotion of Advanced Research (IFCPAR/CEFIPRA), USIAS, FRM and ANR grants, and by the CNRS/University/Inserm IRP Machub. R. Sakr was supported by the French state fund through a doctoral contract from the University of Strasbourg, the Fondation pour la Recherche Médicale (FDT2020010107630) and the ANR. S. Monticelli was funded by IFCPAR, FRC, and ANR. S. Kizhakkenottiyath Shasthadevan was supported by a doctoral contract from the University of Strasbourg in the frame of the CNRS/University/Inserm IRP Machub. G. Zhang was funded by the Chinese Scholarship Council. T. Tabiat was funded by the ANR. This work of the Interdisciplinary Thematic Institute IMCBio, as part of the ITI 2021-2028 program of the University of Strasbourg, CNRS and Inserm, was supported by IdEx Unistra (ANR-10-IDEX-0002), and by SFRI-STRAT'US project (ANR 20-SFRI-0012) and EUR IMCBio (ANR-17-EURE-0023) under the framework of the French Investments for the Future Program.

## Author contributions

**Rosy Sakr**: Conceptualization; Resources; Data curation; Formal analysis; Investigation; Methodology. **Sara Monticelli**: Conceptualization; Supervision; Validation; Investigation; Visualization; Methodology; Writing—original draft; Project administration; Writing—review and editing. **Smrithi Kizhakkenottiyath Shasthadevan**: Validation; Investigation; Methodology. **Claude Delaporte**: Validation; Methodology. **Gege Zhang**: Validation; Writing—review and editing. **Tarek Tabiat**: Validation; Methodology; Writing—review and editing. **Angela Giangrande**: Conceptualization; Formal analysis; Supervision; Funding acquisition; Validation; Investigation; Writing—original draft; Project administration; Writing—review and editing. **Pierre B Cattenoz**: Conceptualization; Resources; Data curation; Formal analysis; Supervision; Funding acquisition; Investigation; Methodology; Writing—original draft; Project administration; Writing—review and editing.

Source data underlying figure panels in this paper may have individual authorship assigned. Where available, figure panel/source data authorship is listed in the following database record: biostudies:S-SCDT-10_1038-S44319-026-00728-1.

## Disclosure and competing interests statement

The authors declare no competing interests.

