## [Peer Review File · EMBO Reports]

NimA promotes cell adhesion at the blood brain barrier of the *Drosophila* nervous system

Rosy Sakr, Sara Monticelli, Smrithi KIZHAKKENOTTIYATH SHASTHADEVAN, Claude Delaporte, Gege Zhang, Tarek Tabiat, Angela Giangrande, and Pierre Cattenoz

Corresponding author(s): Pierre Cattenoz (cattenoz@igbmc.fr) , Angela Giangrande (angela@igbmc.fr)

Review Timeline:

Submission Date:	13th Jan 25
Editorial Decision:	20th Jan 25
Appeal Received:	11th Jun 25
Editorial Decision:	23rd Jul 25
Revision Received:	28th Nov 25
Editorial Decision:	21st Jan 26
Revision Received:	28th Jan 26
Accepted:	9th Feb 26

Editor: Esther Schnapp

Transaction Report:

20th Jan 2025

Dear Dr. Cattenoz, dear Angela,

Thank you for the (re-)submission of your manuscript to EMBO reports. I have now read and discussed it with my colleagues here, and I am sorry to say that we all agree that as it stands, it is not so well suited for us.

We note that your study identifies the glia-specific protein NimA (an uncharacterized transmembrane protein that belongs to the family of the Nimrod scavenger receptors) by comparing transcriptomes of neurons, macrophages and glia cells from *Drosophila*. You show that NimA is expressed in glia cells forming boundaries, that it promotes BBB integrity, septate junction formation, neural stem cell proliferation, and cell aggregation when overexpressed in S2 cells. We recognize that your data indicate that NimA promotes cell adhesion at the blood brain barrier of the *Drosophila* central nervous system.

However, we also note that it remains unclear how NimA does so, or in which cell types exactly it acts. We think that a little more data/insight would be required for the publication of your study by EMBO reports. For example, a rescue experiment that re-expresses NimA in certain types of glia cells in mutant flies and that identifies the glia population that rescues the BBB defects would be a useful addition. If you have such data, I would like to encourage you to resubmit your ms to EMBO reports. In the absence of either a little more insight into how NimA acts, or in which cell types it acts, we think that the manuscript would be a better fit for our sister journal Life Science Alliance (<http://www.life-science-alliance.org/>; our broad scope Open Access journal published in partnership between the EMBO-, Rockefeller University-, and Cold Spring Harbor Laboratory Presses).

Eric Sawey, Executive Editor of Life Science Alliance (e.sawey@life-science-alliance.org) would be pleased to send your manuscript as it stands now for in-depth peer review; no reformatting is required. We very much hope that you will be interested in this option: please follow the link below for transfer.

For EMBO reports, I am sorry that I cannot be more positive this time, and I thank you once more for your interest in our journal.

** As a service to authors, EMBO Press provides authors with the ability to transfer a manuscript that one journal cannot offer to publish to another journal, without the author having to upload the manuscript data again. To transfer your manuscript to another EMBO Press journal using this service, please click on Link Not Available

igbmc
institut de génétique et de
biologie moléculaire et cellulaire

Inserm

Strasbourg, June 11th 2025

Dear Esther,

We are finally resubmitting the manuscript you saw in January:

NimA promotes cell adhesion at the blood brain barrier of *Drosophila* nervous system

by Rosy Sakr, Sara Monticelli, Claude Delaporte, Gege Zhang, Tarek Tabiat, myself and Pierre B. Cattenoz

In your mail of January 20th, you indicated that 'a little more data/insights would be required for the publication of the study in EMBO reports'. We hence performed additional analyses aiming at understanding how NimA controls cell adhesion and in which glial cells it is required.

As per the first issue, we could demonstrate that the blood brain barrier phenotype observed upon mutating NimA depends on its expression in the subperineural glia. We did this upon crossing Gal4 drivers specific to different glial subpopulations (perineural, subperineural, cortex glia) with two independent NimA RNAi transgenic lines.

We also dug into the molecular mechanisms and showed that the Nim domain is necessary for cell adhesion. We constructed a NimA expression vector deleted of the EMI domain and showed that cells transfected with this mutant vector do not aggregate.

We are now resubmitting the manuscript and look forward to your feedback.

Best,

Angela Giangrande and Pierre Cattenoz

Dear Dr. Cattenoz,

Thank you for the submission of your manuscript to EMBO reports. We have now received the full set of referee reports that is pasted below.

As you will see, the referees acknowledge that the findings are potentially interesting. However, together they also raise some concerns and have several suggestions for how the study could be improved and strengthened. I think all suggestions are good and should be addressed, but please let me know in case you disagree and we can discuss the exact revision requirements further, also in a video chat, if you like.

I would thus like to invite you to revise your manuscript with the understanding that the referee concerns must be fully addressed and their suggestions taken on board. Please address all referee concerns in a complete point-by-point response. Acceptance of the manuscript will depend on a positive outcome of a second round of review. It is EMBO reports policy to allow a single round of major revision only and acceptance or rejection of the manuscript will therefore depend on the completeness of your responses included in the next, final version of the manuscript.

We realize that it is difficult to revise to a specific deadline. In the interest of protecting the conceptual advance provided by the work, we recommend a revision within 3 months (23rd Oct 2025). Please discuss the revision progress ahead of this time with the editor if you require more time to complete the revisions.

- 1) A data availability section providing access to data deposited in public databases is missing. If you have not deposited any data, please add a sentence to the data availability section that explains that.
- 2) Your manuscript contains statistics and error bars based on $n=2$. Please use scatter blots in these cases. No statistics should be calculated if $n=2$.

3) We replaced Supplementary Information with Expanded View (EV) Figures and Tables that are collapsible/expandable online. A maximum of 5 EV Figures can be typeset. EV Figures should be cited as 'Figure EV1, Figure EV2' etc... in the text and their respective legends should be included in the main text after the legends of regular figures.

5) a complete author checklist, which you can download from our author guidelines . Please insert information in the checklist that is also reflected in the manuscript. The completed author checklist will also be part of the RPF.

6) Please note that all corresponding authors are required to supply an ORCID ID for their name upon submission of a revised manuscript (. Please find instructions on how to link your ORCID ID to your account in our manuscript tracking system in our Author guidelines

- the name of the statistical test used to generate error bars and P values,
- the number (n) of independent experiments (please specify technical or biological replicates) underlying each data point,
- the nature of the bars and error bars (s.d., s.e.m.),
- If the data are obtained from n {less than or equal to} 2, use scatter blots showing the individual data points.

12) All Materials and Methods need to be described in the main text using our 'Structured Methods' format, which is required for all research articles. According to this format, the Methods section includes a separate Reagents and Tools Table file (listing key reagents, experimental models, software and relevant equipment and including their sources and relevant identifiers) and a Methods and Protocols section describing the methods using a step-by-step protocol format. The aim is to facilitate adoption of the methodologies across labs. More information on how to adhere to this format as well as a downloadable template (.docx) for the Reagents and Tools Table can be found in our author guidelines:

An example of a Method paper with Structured Methods can be found here: <https://www.embopress.org/doi/full/10.1038/s44320-024-00037-6#sec-4>

As part of the EMBO publication's Transparent Editorial Process, EMBO reports publishes online a Review Process File (RPF) to accompany accepted manuscripts. This File will be published in conjunction with your paper and will include the referee

reports, your point-by-point response and all pertinent correspondence relating to the manuscript.

I look forward to seeing a revised form of your manuscript when it is ready.

Referee #1:

In this manuscript the authors focus on the role of the *Drosophila* NimA protein and demonstrate that NimA can act as an adhesion protein in cell culture and is required in the blood-brain barrier of *Drosophila*.

In their brief paper, the authors first summarize the results of bulk mRNA sequencing to identify genes expressed in neurons, glial cells, and hemocytes. They selected NimA from the list of 129 genes; NimA belongs to the Nimrod family of scavenger receptors.

In a first set of experiments, the authors demonstrate that NimA likely does not act as a scavenger receptor in S2 cells. Furthermore, NimA—the fly ortholog of the human platelet endothelial aggregation receptor 1 (PEAR1)—rather provides adhesive functions, as determined in S2 cells and in vivo. Unfortunately, the different images of S2 cells demonstrating phagocytosis inefficiency and cell aggregation are inconsistent in the paper. As no aggregation can be detected in the phagocytosis. This should be consistent and at least addressed in the discussion.

To address the role of NimA the authors used a CRISPR/Cas9 to replace the entire NimA gene with the Gal4 coding sequence. These null mutants are homozygous viable and show no defects in PDF-tri neuron removal.

This CRISPR/Cas9 based replacement strategy also removes all intronic sequences that may harbor relevant regulatory sequences, so it remains unclear whether this reporter faithfully reflects the normal NimA expression pattern. In the larval CNS, this driver directs expression in ensheathing glia and blood brain barrier forming glia of the brain lobes. The level of resolution of the images does not allow to judge whether expression is found in the perineurial or the subperineurial glia or in both glia subtypes. This needs to be documented much better, in particular activity in the adult blood brain barrier should be demonstrated.

There is a MiMIC insertion available which can be easily used to generate a Trojan Gal4 insertion that will (a) generate a null allele and (b) will give information on the expression of the NimA mRNA. In addition a GFP tag could be inserted to determine the (sub)cellular localization of NimA in glia. This is in particular relevant to support the interesting finding that NimA mutant brains show disruptions in the organization of the occluding septate junctions. Is this observed only in the brain lobes where in the larva NimA expression is expected? How is the septate junction phenotype in the vnc or in adult stages (where no NimA is detected in the surface glia).

Finally, the mutant phenotype is observed in homozygous NimAGal4 knockin animals. Is the phenotype present in animals carrying the NimAGal4 allele in trans to a deficiency? CRISPR/Cas9 is known for off target effects and the septate junction phenotype or the small brain size phenotype could be due to such an off target. What is crucially important is a rescue experiment which should be easily possible given that UAS-NimA transgenes are already generated. Such rescue experiments could also allow to answer the question whether the brain lobe reduction phenotype is due to missing NimA function in the blood brain barrier or due to defective ensheathing glia.

Minor points

The list of 129 glial genes as well as the 431 neuronal and the 625 macrophage specific genes should be provided more clearly in different tables. In the supplementary table 134 genes are listed as glia specific.

The recent transcriptomic analyses by the Fernandes lab should be mentioned and discussed here.

It should be perineurial and subperineurial and not perineurial and subperineurial.

Typo in line 236 "Specie".

Referee #2:

The manuscript by Sakr et al., "NimA promotes cell adhesion at the blood brain barrier of *Drosophila* nervous system," explores the role of NimA in glial cells as essential for normal brain development. The authors identify NimA as a pan-glial marker with

greater specificity than Repo in larva, and they provide compelling evidence that NimA contributes to BBB integrity by influencing septate junctions in the subperineurial glia (SPG). These findings are novel and of clear interest to the readership of EMBO Reports, contributing to our understanding of glial identity and BBB development.

Strengths: The authors identify NimA as a pan-glial marker which can distinguish between glia from macrophages at L3 even better than Repo, an established marker of glial cells. The authors nicely demonstrate that NimA has a role in formation or maintaining BBB permeability via changes to the septate junction in the SPG. They also define a role for NimA in cell-cell adhesion in S2 cells.

Weaknesses: The conclusion that NimA does not play a role in phagocytosis is inadequately supported. The role of NimA in EC/CG or in the adult fly is unclear. These points are detailed below.

This manuscript makes a significant contribution to the field of glial biology and BBB development in *Drosophila*. With some clarification regarding NimA's role in phagocytosis and EC/CG and adult glia, it will be suitable for publication in EMBO Reports.

Comments:

While bead uptake in S2 cells and tri-neuron clearance in adults were used to argue against a phagocytic role, these assays may not provide the appropriate biological context. S2 cells, though of hemocyte origin, do not express NimA and likely lack the necessary machinery for its function. Similarly, the tri-neuron model in adults may not be sensitive enough to detect a subtle or context-dependent role in glial phagocytosis. Additional experiments in larval models of injury, where phagocytic responses of cortex and ensheathing glia are more robust and well-characterized, could more effectively address this question. Considering the scRNA-seq data and antibody staining show that the adult flies only express NimA in EG/CGs, what role, if any, do you think NimA has in adult glia?

Are all the effects of the NimA (increased BBB permeability, developmental delay) due to its role in development or is NimA involved in adult maintenance of permeability?

For example, if you knockdown NimA in after L3, would you still get increased Dextran permeability?

Referee #3:

This manuscript by Sakr et al reports the novel finding that NimA, a member of the Nimrod scavenger receptor family, is involved in the formation of the glial blood brain barrier (BBB) in *Drosophila* instead. Importantly, this novel function correlates with the loss of extracellular EGF-like repeats and high evolutionary conservation of NimA. The data clearly show that NimA over-expression does not promote phagocytosis of latex beads in cell culture, that NimA is expressed in glial cells, it is required for the integrity of the BBB in vivo, and can promote cell aggregation in culture. Overall, the data are of good quality (e.g, the parallel expression of NimA and reporters using T2A is very nice, enables untagged expression of proteins of interest and replaces the second ribosome entry site methods which do not always work so well) and the microscopy is beautiful. Altogether, these are interesting findings that will contribute to the understanding of glial cell biology and the evolutionary paths between immunity and nervous system functions.

Please find below suggestions that would help improve the work.

1. The current version does not explain the rationale and methodology for assigning gene expression data to distinct cell types, from data presented in Figure 1A-C and Supplementary Figure S1. This diminishes the validity of the claim that NimA is specifically expressed in glia. This is a very important point, as the main claim of the paper is that although NimA is a member of the scavenger protein family, unexpectedly, it does not have immune functions and it is instead involved in glial-to-glia cell adhesion to form the BBB.

The authors say the claim is based on bulk RNAseq data, but if this is so, this method does not have the resolution to assign transcripts to distinct cell types, and GO terms (eg Figure 1B) are not reliable means to identify cell types. The Methods section does not describe how the data were obtained. They say the data in Figure 1 and Figure S1 is an analysis of previously published data by these same authors in Cattenoz et al (2020) EMBO J and findings currently in bioRxiv with a different manuscript title (Sakr et al 2022). Here (2022), they say that NimA is expressed in both glia and hemocytes. To enable the data analysed in Figure 1 and Figure S1 to support the claim that NimA is expressed in glia, the authors should provide detailed information of how the RNAseq was carried out and how distinct cell types were identified (i.e. was it bulk RNAseq from intact embryos and from dissected L3 CNS, as mentioned in this manuscript, or bulk RNAseq from purified glial cells, hemocytes and neurons, as mentioned in the above references?).

It would also help to clarify the relationship of this manuscript with the Sakr 2022 bioRxiv: 2022.06.30.498263 (i.e. is it currently under consideration in a different journal? Or is the manuscript under assessment a new version of the same work?).

2. The authors claim that Figure 1F-J shows that over-expressed NimA does not induce the phagocytosis of latex beads compared to controls. However, there is no positive control. This is an important point, as a key claim is that NimA does not promote phagocytosis, and that in evolution NimA diverged away from its paralogues to lose immune functions and take on cell-

adhesion functions. The data and reliability of the claim would improve with a positive control that demonstrates that an over-expressed scavenger receptor is attracted to phagocytose latex beads. Alternatively, they could induce cell death (e.g. by co-expressing reaper, hid or grim) together with over-expressing NimA and visualise anti-Dcp1 or DAPI (as apoptosis induces nuclear condensation and fragmentation) together with NimA.

3. Point 3.1 Figure 2 data are used to claim that loss of NimA does not alter the course of PDF-tri neuron elimination at 3 days after adult eclosion and therefore NimA function is dispensable for the clearance of PDF-tri neurons. However, G,H,I seem to show fewer PDF+ spots than controls, which would suggest that in NimA mutants clearance is faster or more effective. Could this suggest that NimA can bind other family members (e.g. Drpr) to reduce their function (i.e. dominant negative or sink effect?), and this is lost in the mutants? Either way, to strengthen or refute their claim, the authors ought to quantify PDF+ spots (if they already use Imaris they could use the spot function) or signal intensity (with Imaris in volume or with Fiji in projections). Furthermore, Figure 2L seems to have a break in the axons. Could this reveal a function of NimA in enabling contact between these PDF branches? This phenotype should be quantified.

Point 3.2 As NimAGal4-KO/+ is a heterozygous mutant, it is not a control. The use of NimAGal4 to visualise the expression of NimA is good to show NimA is located at the right place and time. Additionally, as control the authors should show the levels of PDF in wild-type.

4. Figure 2G-M and Figure 3E-M use homozygous NimA-KO-T2A-Gal4 specimens as null mutants. However, it is well-known that the use of homozygous mutants will also show the contribution of the genetic background to the phenotypes. The authors state in the text that there are other NimA null mutant alleles available. To claim functions of NimA based on LOF conditions, the authors must use trans-heterozygous combinations of two alleles (ie one allele over the other), to eliminate the contribution to the phenotype of the genetic background - as it is standard in the field.

5. Figure 4: The authors claim that loss of function for NimA reduces overall CNS size, and that this does not reflect a function for NimA in promoting apoptosis, but in reducing proliferation and Dpn+ cell number instead. However, these claims are not well supported by the data. The Methods section does not explain how larval staging was carried out. Figure 4G-O show that NimA LOF causes developmental delay, with homozygote NimA mutant larvae pupariating around one day later than heterozygotes (wild-type controls are missing and must be included). This delay would cause a 96h old larva to look like a 72h old larva, which are considerably thinner. As a consequence, the extent of proliferation and Dpn+ cell number would correspond to that of younger larvae, invalidating their claim. To solve this problem, the authors could measure brain size, cell proliferation and Dpn+ cell number in the mutants 24h later than in wild-type controls (if 24h is the period by which mutants are delayed, otherwise adjust accordingly) - or explain whether in fact this is what they did. To measure the incidence of cell death, Dcp1 must be analysed before the onset of an overt phenotype of decreased CNS size. That is, if the CNS is already smaller, then cell death already happened and will no longer be detectable. Apoptosis and the clearance of dead cells are both fast processes, thus anti-Dcp1+ cells are detectable within a narrow time window. This means that to quantify Dcp1+ cells, accurate staging needs to be used in order to obtain reliable data. Additionally, it can be challenging to identify the timing of cell death. The authors must repeat these experiments before the CNSs become overtly smaller, or rescue cell death e.g. with p35, as an alternative approach, or remove the claim.

Minor points:

Figure 5M should be mentioned within the results section, rather than only in the Discussion.

Line 218 page 8, Discussion: "Our transcriptomic approach identifies a panel of 129 genes constitutive of a robust molecular signature of glia". As explained above, not sufficient evidence and information have been provided in this manuscript to support this statement. See above, point 1.

Line 243-244, page 9 Discussion: "The high conservation of the NimA extracellular domain also suggests specific conserved ligands activating the receptor." However, Figures 3 and 5 suggest a function for NimA in homophilic cell adhesion - which would not necessitate any other ligand.

Line 250-252, page 9 Discussion: "page 9 Discussion: "Since several conserved tyrosine residues are also predicted to be phosphorylated in the intracellular domain of NimA (Figure 5M), it is possible that its multimerization leads to tyrosine phosphorylation to activate the downstream cascade responsible for cell aggregation". Wouldn't cell-to-cell adhesion (i.e. at the PM) suffice?

Figure 3 is very nice. Figure 3B shows NimA to be expressed in surface glia/BBB surrounding the optic lobes but not the VNC. Why not?

Similarly, the breakdown of BBB affects optic lobes in Figure 3E-F. Does it also occur in the VNC?

Regarding Figure 3I, It would be nice to see the same magnification (ie whole CNS) for the NimA-KOGal4 mutant too. Does the phenotype in K happen also in the VNC? If not, could they speculate why?

Figure 3C-D are not mentioned within the text.

Dear Esther,

We thank you and the reviewers for the careful evaluation of our manuscript. The comments have helped us to improve the quality and clarity of our work upon performing a number of new experiments and analyses to address their concerns.

Below, we provide a detailed, point-by-point response to each of the reviewers' comments. Changes made are highlighted in yellow in the manuscript.

Referee #1

In this manuscript the authors focus on the role of the *Drosophila* NimA protein and demonstrate that NimA can act as an adhesion protein in cell culture and is required in the blood-brain barrier of *Drosophila*.

In their brief paper, the authors first summarize the results of bulk mRNA sequencing to identify genes expressed in neurons, glial cells, and hemocytes. They selected NimA from the list of 129 genes; NimA belongs to the Nimrod family of scavenger receptors.

In a first set of experiments, the authors demonstrate that NimA likely does not act as a scavenger receptor in S2 cells. Furthermore, NimA-the fly ortholog of the human platelet endothelial aggregation receptor 1 (PEAR1)-rather provides adhesive functions, as determined in S2 cells and *in vivo*.

We thank referee #1 for the attention to detail and the valuable suggestions.

1) Unfortunately, the different images of S2 cells demonstrating phagocytosis inefficiency and cell aggregation are inconsistent in the paper. As no aggregation can be detected in the phagocytosis. This should be consistent and at least addressed in the discussion.

To address the reviewer's comment, we now provide a more direct and biologically relevant readout. We replaced the bead uptake assay in S2 cells with two independent *in vivo* assays probing NimA involvement in phagocytosis during development.

The absence of large cell clumps in the original images illustrating the phagocytic ability of S2 cells overexpressing NimA was due to the incomplete penetrance of the phenotype. Even upon NimA overexpression, most cells remain as singlets or form only small aggregates of 2–4 cells, as quantified in the **Figure 5F**. Moreover, to quantify the phagocytic potential in a reliable manner, we actually focused on single cells. Hence the bias in the presented data.

2) To address the role of NimA the authors used a CRISPR/Cas9 to replace the entire NimA gene with the Gal4 coding sequence. These null mutants are homozygous viable and show no defects in PDF-tri neuron removal. This CRISPR/Cas9 based replacement strategy also removes all intronic sequences that may harbor relevant regulatory sequences, so it remains unclear whether this reporter faithfully reflects the normal NimA expression pattern.

We understand the issue raised by the reviewer. In the original version of the manuscript, we included results from an *in situ* hybridization assay showing *NimA* transcripts in the wandering L3 (wL3) CNS (**Supplementary Figure S3A–A'''** and shown below). However, no

additional glial markers were included at that time.

To refine the analysis, we have now performed multiple *in situ* labeling experiments targeting *NimA* together with *Repo* as a pan-glial marker. This experiment validates the previous *in situ* results as well as the *NimA*-Gal4–derived expression data. As shown in the revised **Figure EV2G–H'''** and below, *NimA* transcripts are present in surface, cortex, and

ensheathing glia, where they colocalize with *Repo*.

3) In the larval CNS, this driver directs expression in ensheathing glia and blood brain barrier forming glia of the brain lobes. The level of resolution of the images does not allow to judge whether expression is found in the perineurial or the subperineurial glia or in both glia subtypes. This needs to be documented much better, in particular activity in the adult blood brain barrier should be demonstrated.

To resolve the identity of the surface glial subtype expressing *NimA*, we combined *in situ* hybridization assays targeting *NimA* with the subperineurial-specific marker *Moody*. This experiment, included in **Figure 3C–D'''** and shown below, allows clear discrimination between perineurial and subperineurial glia and proves that *NimA* is expressed in subperineurial but not in perineurial glia.

Concerning the adult BBB, our results show that loss of NimA does not cause additional developmental delay or lethality beyond the larval stage, and adult lifespan is comparable to controls (**Figure 4G,K, Figure EV4A-C**). Consistently, NimA knockdown in subperineurial glia produced similar outcomes (**Figure 4H,L, Figure EV4D,G**). This indicates that NimA acts primarily during development, with adult homeostasis likely being maintained by compensatory mechanisms. This also aligns with the very low NimA expression detected in adult subperineurial glia in scRNA-seq datasets (**Figure 2K,K'** and figure below) and in our NimA KO-Gal4 reporter line (figure below). As NimA function in the BBB seems to be most prominent during larval development, our phenotypic analyses focus on this stage.

Single confocal section, adult brain *NimA KO-Gal4/+;UAS-mCD8-RFP*

- 4) There is a MiMIC insertion available which can be easily used to generate a Trojan Gal4

insertion that will (a) generate a null allele and (b) will give information on the expression of the NimA mRNA. In addition a GFP tag could be inserted to determine the (sub)cellular localization on NimA in glia.

We thank the reviewer for this suggestion. We are aware of the MiMIC insertion (MiMICNimAMI11280) and agree that the Trojan-GAL4 insertion could provide both a null allele and an expression reporter. However, this approach would functionally overlap with the CRISPR/Cas9-based gene replacement strategy that we have now validated by *in situ* hybridization, and for which we have already characterized the loss-of-function phenotypes.

The localization of NimA is an important issue. To address this, we commissioned WellGenetics (Taiwan) to generate a fly line in which the endogenous NimA is tagged with MYC using CRISPR/Cas9. Compared to GFP, the MYC tag is much smaller and is generally not expected to interfere with protein trafficking/function. Unfortunately, although several immunolabeling protocols were used, we are unable to detect a clear MYC signal in the central nervous system, Even though we could see a band corresponding to Nima by Western Blot. In addition, and as an alternative strategy, we generated a custom antibody against NimA. Although the antigen peptide was recommended by the manufacturer (ProteoGenix) as highly likely to generate a detectable antibody response, we were unable to obtain a specific signal for the NimA protein. This could reflect limited accessibility of the chosen epitope in the protein's native structure.

We recognize that the precise localization of NimA remains to be fully resolved. However, we have invested substantial effort and resources in generating multiple tools to address this question. We hope that the reviewer will find the additional functional assays included in the revised manuscript sufficiently compelling to support the presence and relevance of the NimA protein, even though direct detection was not achieved despite these extensive attempts.

5) This is in particular relevant to support the interesting finding that NimA mutant brains show disruptions in the organization of the occluding septate junctions. Is this observed only in the brain lobes where in the larva NimA expression is expected? How is the septate junction phenotype in the vnc or in adult stages (where no NimA is detected in the surface glia).

Our results clearly indicate that NimA functions during development to support BBB integrity and CNS formation, whereas they do not reveal any obvious role for NimA in the adult stage, as discussed in our response to comment 3. Moreover, analyzing BBB structure and permeability in the adult central nervous system is particularly challenging. This is reflected in the limited number of studies addressing adult BBB permeability (Bainton et al., 2005; Li et al., 2021) compared to the extensive work conducted on the embryonic and larval stages, which includes Baumgartner et al., 1996; Petri et al., 2019; Stork et al., 2008; Syed et al., 2011; Schwabe et al., 2005; Ariss et al., 2020; Nelson et al., 2010.

To address the reviewer's question, we also quantified BBB permeability in the VNC by measuring the penetration of 10 kDa dextran in a subset of CNS samples and we did not detect any difference relative to controls (see chart below). Despite detectable (see image below), NimA expression is much lower in the VNC than in the brain lobes (**Figure 1G'**), which is why we initially focused our permeability analyses on the brain lobes as a more sensitive readout of NimA loss of function. Altogether, these results suggest that additional factors, potentially acting together with NimA, help maintain BBB integrity in the VNC. If the reviewer wishes, these data can be included in the manuscript.

6) Finally, the mutant phenotype is observed in homozygous NimAGal4 knockin animals. Is the phenotype present in animals carrying the NimAGal4 allele in trans to a deficiency? CRISPR/Cas9 is known for off target effects and the septate junction phenotype or the small brain size phenotype could be due to such an off target. What is crucially important is a rescue experiment which should be easily possible given that UAS-NimA transgenes are already generated. Such rescue experiments could also allow to answer the question whether the brain lobe reduction phenotype is due to missing NimA function in the blood brain barrier or due to defective ensheathing glia.

The reviewer raises two important points: (1) whether the phenotypes observed in the CRISPR/Cas9 *NimA* mutant could be due to off-target effects, and (2) whether NimA is required cell-autonomously in specific glial subtypes. The only experimental strategy that allows us to address both issues simultaneously and efficiently is a conditional knockdown of NimA, which we have now expanded.

The characterized *NimA* mutant alleles (which, like ours, display the “jaunty” curved-wing phenotype) carry only point mutations or very small deletions. As a result, they are likely to produce truncated proteins possibly retaining partial function. In contrast, our CRISPR/Cas9 *NimA* mutant is a true null, as the entire coding sequence has been removed. Because we do not know which domains of NimA are required for BBB integrity or CNS development, trans-heterozygous combinations using possibly hypomorphic alleles could produce false-negative results simply because the critical domains remain intact in those backgrounds. This consideration is further supported by our observation that silencing NimA in specific glial subtypes (ensheathing, cortex, or subperineurial glia) does not reproduce the jaunty wing phenotype, indicating that the jaunty phenotype is not necessarily linked to NimA’s role in BBB formation or CNS development.

To rule out the possibility that the BBB phenotypes observed in the CRISPR/Cas9 *NimA* mutant arise from off-target effects, while simultaneously determining the glial subtype in which NimA is required, we expressed a NimA RNAi line in distinct glial populations using established drivers for subperineurial glia (*Mdr65-GAL4*, Mayer et al., 2009; Pfeiffer et al., 2010), perineurial glia (*shn-GAL4*, Pfeiffer et al., 2010; Kozlov et al., 2020), and cortex glia (*VGlut2-GAL4*, Pfeiffer et al., 2010; Kremer et al., 2017) and we analyzed the BBB phenotypes. Only the knockdown in subperineurial glia recapitulates the increased BBB

permeability observed in the *NimA* mutant. (Figure 4H, data already present in the manuscript), similar to that observed in the mutant. Moreover, we confirmed the increased BBB permeability using a second independent RNAi line, supporting that the phenotype is not due to CRISPR/Cas9 off-target effects and demonstrating a specific requirement for *NimA* in subperineurial glia for BBB integrity. Finally, the increased BBB permeability specifically in the brain lobes, where *NimA* expression is higher than in the VNC (see response to the comment 5), further supports that this phenotype results from the loss of *NimA*, rather than from potential CRISPR/Cas9 off-target effects.

To determine whether the CNS defects (reduced brain volume, proliferation, and neuroblast number) observed in the *NimA* mutant (Figure 4A-E) reflect *NimA* function in subperineurial glia and/or in other glial populations where *NimA* is expressed (cortex and ensheathing glia) we now extended our analysis upon knocking down *NimA* in each of these glia subtypes. The results are included in Figure 4M-O and shown below. Consistent with the known importance of an intact BBB for neuroblast proliferation and CNS growth, subperineurial glia-specific *NimA* knockdown leads to CNS defects comparable to those of the *NimA* mutant. Interestingly, similar phenotypes are also observed when *NimA* is knocked down in cortex glia, and this is also in line with the established role of this glia subtype as a relay in the signal regulating neuroblast proliferation (Rujano et al., 2022; Bailey et al., 2015; Morante et al., 2013; Plazaola-Sasieta et al., 2019). No evident developmental phenotype was observed upon downregulation of *NimA* in ensheathing glia. Overall, these data demonstrate that *NimA* is required in subperineurial glia for BBB integrity, and, together with its role in cortex glia, for neuroblast support and CNS growth.

list of 129 glial genes as well as the 438 neuronal and the 629 macrophage specific genes should be provided more clearly in different tables. In the supplementary table 134 genes are listed as glia specific.

The lists of marker genes have been appended as additional sheets in Table EV1. We also corrected the number of marker genes. We thank the referee for spotting this mistake.

8) The recent transcriptomic analyses by the Fernandes lab should be mentioned and discussed here.

Following the reviewer suggestion, we analyzed NimA expression in Fernandes data (Lago-Baldaia et al., 2023, PLOS Biology). NimA was not found enriched in any glia cluster defined in this study and we could observe only low expression in limited number of chiasm, EG and CG (see figure A below). The discrepancy with the data from Davies et al. (2018) that show clear enrichment in EG and CG (**Figure 2K,K'**), is explained by the difference in sequencing depth. The datasets used by Lago-Baldaia et al. have a median sequencing depth of 1852 transcripts per cells (see the distribution in figure B below) to compare to 53553 transcripts per cell in Davies et al. dataset.

Thus, the Fernandes dataset does not provide additional functional insight relevant to our study. We therefore did not include this study in the manuscript.

9) It should be perineurial and subperineurial and not perineurial and subperineurial.

We apologize for the oversight, which has been corrected in the revised manuscript.

10) Typo in line 236 "Specie".

We apologize for the oversight, the typo has been corrected.

Referee #2

The manuscript by Sakr et al., "NimA promotes cell adhesion at the blood brain barrier of *Drosophila* nervous system," explores the role of NimA in glial cells as essential for normal brain development. The authors identify NimA as a pan-glial marker with greater specificity than Repo in larva, and they provide compelling evidence that NimA contributes to BBB integrity by influencing septate junctions in the subperineurial glia (SPG). These findings are novel and of clear interest to the readership of EMBO Reports, contributing to our understanding of glial identity and BBB development.

Strengths: The authors identify NimA as a pan-glial marker which can distinguish between glia from macrophages at L3 even better than Repo, an established marker of glial cells. The authors nicely demonstrate that NimA has a role in formation or maintaining BBB

permeability via changes to the septate junction in the SPG. They also define a role for NimA in cell-cell adhesion in S2 cells.

Weaknesses: The conclusion that NimA does not play a role in phagocytosis is inadequately supported. The role of NimA in EC/CG or in the adult fly is unclear. These points are detailed below.

This manuscript makes a significant contribution to the field of glial biology and BBB development in *Drosophila*. With some clarification regarding NimA's role in phagocytosis and EC/CG and adult glia, it will be suitable for publication in EMBO Reports.

We appreciate referee # 2 's positive reception and suggestions, which were all taken into account.

Comments:

1) While bead uptake in S2 cells and tri-neuron clearance in adults were used to argue against a phagocytic role, these assays may not provide the appropriate biological context. S2 cells, though of hemocyte origin, do not express NimA and likely lack the necessary machinery for its function. Similarly, the tri-neuron model in adults may not be sensitive enough to detect a subtle or context-dependent role in glial phagocytosis. Additional experiments in larval models of injury, where phagocytic responses of cortex and ensheathing glia are more robust and well-characterized, could more effectively address this question.

We agree with the reviewer that S2 cells may lack essential components required for NimA-mediated phagocytosis, as they do not express NimA. However, larval injury models are technically challenging and require extensive routine practice, expertise that our laboratory does not currently have, in order to minimize variability and obtain reliable results. Therefore, to address the reviewer's comment, we replaced the S2 cell bead uptake assay with an alternative *in vivo* phagocytic assay that examines the clearance of Corazonin-expressing neurons (vCrz) in the ventral nerve cord during early pupal development. The vCrz neurons consist of eight pairs along the larval ventral nerve cord, undergo apoptosis, and are normally removed by the surrounding EG and CG within ~6 h after puparium formation (APF) (Choi et al., 2006; Tasdemir-Yilmaz & Freeman, 2014; Perron, Carme et al., 2023). Importantly, these glial cells express NimA (see images below), making this system well suited to assess the role of NimA in phagocytosis in a physiological context.

We found that vCrz neurons are cleared with similar efficiency in wild-type, *NimA KO-Gal4/+* and *NimA KO-Gal4* animals at 4 h APF, indicating that loss of NimA does not impair the phagocytic removal of vCrz neuron cell bodies. This experiment is included in **Figure 2A-I** and shown below.

Central nervous system from wandering larvae (WL3) and pupae (4 h after puparium formation, hAPF)
 Control (*UAS-mCD8-RFP*), *NimA KO-Gal4/+* (*NimA-T2A-Gal4/+;UAS-mCD8-RFP*) and *NimA KO-Gal4* (*NimA-T2A-Gal4;UAS-mCD8-RFP*).

Moreover, we confirmed these findings with an independent assay, the cleaved Death caspase-1 (Dcp-1) immunofluorescence assay, previously established to identify uncleared apoptotic debris in the third-instar larval nervous system (McLaughlin et al., *Dev Cell*, 2019). We quantified Dcp-1 signal in wandering L3 CNS upon *NimA* downregulation in either ensheathing or cortex glia. In either condition, we observed no significant difference compared to controls (data included in **Figure 2J** and shown below), consistent with the vCrz and PDF-neuron clearance assays. Altogether, these results support the conclusion that *NimA* is not likely to play a role in glial phagocytosis.

- 2) Considering the scRNA-seq data and antibody staining show that the adult flies only express NimA in EG/CGs, what role, if any, do you think NimA has in adult glia?

This is an interesting point, and we thank the reviewer for raising it. Based on the new experiments, NimA appears to be required during the larval stage in cortex glia, but not in ensheathing glia, to support CNS and overall animal development (**Figure 4I,J,L-O**). However, this requirement seems restricted to development: downregulation of NimA in cortex glia (or in ensheathing glia) does not affect adult viability, as reflected by normal eclosion rates and the absence of lethality (**Figure EV4E-G**). Consistently, *NimA* mutants exhibit similar phenotypes and display a normal lifespan (**Figure EV4A-C**). Furthermore, glial phagocytosis appears preserved in these mutants, as demonstrated by two independent phagocytic assays performed at pupal (**Figure 2A-I**) and adult stage (**Figure 2M-V'**). Although we cannot exclude that NimA also play a role in adult cortex and ensheathing glia, that would require more extensive analysis, which we consider beyond the scope of the current manuscript.

- 3) Are all the effects of the NimA (increased BBB permeability, developmental delay) due

to its role in development or is NimA involved in adult maintenance of permeability?

For example, if you knockdown NimA in after L3, would you still get increased Dextran permeability?

Our results indicate that NimA functions primarily during larval development. At this stage, NimA is essential in subperineurial glia to ensure proper BBB sealing (**Figure 3H**). The developmental delay observed at pupariation in *NimA* mutants as well as upon NimA downregulation in subperineurial glia (**Figure 4H**) is consistent with its developmental requirement. The absence of additional delay or lethality during eclosion (**Figure EV4D,G**) suggests the presence of compensatory mechanisms that restore homeostasis in the adult brain, where subperineurial glia barely express NimA (see response to comment 3 from Referee #1). Such mechanisms are likely to intervene to mitigate the effect of NimA loss at adult stage. Based on this, and given the technical challenges associated with analyzing the adult BBB (see response to comment 5 from Referee #1), we did not pursue this experiment.

Referee #3

This manuscript by Sakr et al reports the novel finding that NimA, a member of the Nimrod scavenger receptor family, is involved in the formation of the glial blood brain barrier (BBB) in *Drosophila* instead. Importantly, this novel function correlates with the loss of extracellular EGF-like repeats and high evolutionary conservation of NimA. The data clearly show that NimA over-expression does not promote phagocytosis of latex beads in cell culture, that NimA is expressed in glial cells, it is required for the integrity of the BBB in vivo, and can promote cell aggregation in culture. Overall, the data are of good quality (e.g, the parallel expression of NimA and reporters using T2A is very nice, enables untagged expression of proteins of interest and replaces the second ribosome entry site methods which do not always work so well) and the microscopy is beautiful. Altogether, these are interesting findings that will contribute to the understanding of glial cell biology and the evolutionary paths between immunity and nervous system functions. Please find below suggestions that would help improve the work.

We are grateful to Referee #3 for appreciating the significance of our findings and for the suggestions.

1) The current version does not explain the rationale and methodology for assigning gene

expression data to distinct cell types, from data presented in Figure 1A-C and Supplementary Figure S1. This diminishes the validity of the claim that NimA is specifically expressed in glia. This is a very important point, as the main claim of the paper is that although NimA is a member of the scavenger protein family, unexpectedly, it does not have immune functions and it is instead involved in glial-to-glia cell adhesion to form the BBB.

The authors say the claim is based on bulk RNAseq data, but if this is so, this method does not have the resolution to assign transcripts to distinct cell types, and GO terms (eg Figure 1B) are not reliable means to identify cell types. The Methods section does not describe how the data were obtained. They say the data in Figure 1 and Figure S1 is an analysis of previously published data by these same authors in Cattenoz et al (2020) EMBO J and findings currently in bioRxiv with a different manuscript title (Sakr et al 2022). Here (2022), they say that NimA is expressed in both glia and hemocytes. To enable the data analysed in Figure 1 and Figure S1 to support the claim that NimA is expressed in glia, the authors should provide detailed information of how the RNAseq was carried out and how distinct cell types were identified (i.e. was it bulk RNAseq from intact embryos and from dissected L3 CNS, as mentioned in this manuscript, or bulk RNAseq from purified glial cells, hemocytes and neurons, as mentioned in the above references?).

It would also help to clarify the relationship of this manuscript with the Sakr 2022 bioRxiv: 2022.06.30.498263 (i.e. is it currently under consideration in a different journal? Or is the manuscript under assessment a new version of the same work?).

We apologize for the lack of clarity. To identify glia-specific genes, we performed a bulk RNA sequencing assay on glia, neurons and macrophages that were FACS-sorted from whole embryos (E16 glia, E16 neuron and E16 macrophage datasets), wL3 hemolymph (wL3 macrophage dataset) or wL3 central nervous systems (wL3 glia and wL3 neuron datasets) using specific markers (*elav-nRFP*: pan-neural marker, *repo-nRFP*: pan-glial marker, *srp(hemo)Gal4/+*; *UAS-RFP/+* : embryonic macrophage marker, *HmlΔ-RFP/+* : larval macrophage marker). We have clarified this experimental strategy in the revised manuscript, both in the Results section (**lines 67–70**) and in the Methods (**line 349-384**).

Regarding the relationship of this manuscript with *Sakr et al., 2022* (bioRxiv: 2022.06.30.498263): the findings included in that reference were divided into two independent manuscripts addressing distinct questions, following the suggestion of reviewers. The current manuscript (EMBOR-2025-61159V2-Q) focuses on the role of the glia-specific receptor NimA. A second manuscript focusing on a distinct issue is in preparation. In the preprint (*Sakr et al., 2022* bioRxiv: 2022.06.30.498263), NimA is described as glia-specific, consistently with qPCR data showing no expression in L3 macrophages (Supplementary Figure 3 of the preprint).

2) The authors claim that Figure 1F-J shows that over-expressed NimA does not induce the phagocytosis of latex beads compared to controls. However, there is no positive control. This is an important point, as a key claim is that NimA does not promote phagocytosis, and

that in evolution NimA diverged away from its paralogues to lose immune functions and take on cell-adhesion functions. The data and reliability of the claim would improve with a positive control that demonstrates that an over-expressed scavenger receptor is attracted to phagocytose latex beads. Alternatively, they could induce cell death (e.g. by co-expressing reaper, hid or grim) together with over-expressing NimA and visualise anti-Dcp1 or DAPI (as apoptosis induces nuclear condensation and fragmentation) together with NimA.

We thank the reviewer for this comment. In the revised manuscript, the bead uptake assay in S2 cells has been replaced with two independent *in vivo* assays probing NimA involvement in phagocytosis during development (see response to comment 1 from Referee #2). This provides a more direct and biologically relevant readout than the previous *in vitro* experiments.

3) Figure 2 data are used to claim that loss of NimA does not alter the course of PDF-tri neuron elimination at 3 days after adult eclosion and therefore NimA function is dispensable for the clearance of PDF-tri neurons. However, G,H,I seem to show fewer PDF+ spots than controls, which would suggest that in NimA mutants clearance is faster or more effective. Could this suggest that NimA can bind other family members (e.g. Drpr) to reduce their function (i.e. dominant negative or sink effect?), and this is lost in the mutants? Either way, to strengthen or refute their claim, the authors ought to quantify PDF+ spots (if they already use Imaris they could use the spot function) or signal intensity (with Imaris in volume or with Fiji in projections).

The quantification of PDF spots in NimA adult brain did not show significant differences between the controls (*NimA KO-Gal4/+* and wild-type) and the mutant (*NimA KO-Gal4*). In the three genotypes, most PDF neurons are cleared in 3 days old adults, similarly to what observed in wild type animals (Perron et al. 2023). The data are now included in **Figure EV3D-J** and shown below.

Furthermore, Figure 2L seems to have a break in the axons. Could this reveal a function of NimA in enabling contact between these PDF branches? This phenotype should be quantified.

This is a z-stack artefact. Since the z-stacks are centered on the punctae, they do not always cover completely the PDF horizontal axons that are not in the same frames than the punctae. Please, find below image stacks centered on the transversal neurons showing no defects (these are other images than the stacks shown in **Figure 2M-V'**).

As NimAGal4-KO/+ is a heterozygous mutant, it is not a control. The use of NimAGal4 to visualise the expression of NimA is good to show NimA is located at the right place and time. Additionally, as control the authors should show the levels of PDF in wild-type.

A control genotype was added for the quantification of PDF-neurons (see above and **Figure EV3D-J**).

4) Figure 2G-M and Figure 3E-M use homozygous NimA-KO-T2A-Gal4 specimens as null mutants. However, it is well-known that the use of homozygous mutants will also show the contribution of the genetic background to the phenotypes. The authors state in the text that there are other NimA null mutant alleles available. To claim functions of NimA based on LOF conditions, the authors must use trans-heterozygous combinations of two alleles (ie one allele over the other), to eliminate the contribution to the phenotype of the genetic background - as it is standard in the field.

We appreciate the comment on the potential effects from the genetic background. A detailed response to this point is provided in our reply to comment 6 from Referee #1.

5) Figure 4: The authors claim that loss of function for NimA reduces overall CNS size, and

that this does not reflect a function for NimA in promoting apoptosis, but in reducing proliferation and Dpn+ cell number instead. However, these claims are not well supported by the data. The Methods section does not explain how larval staging was carried out. Figure 4G-O show that NimA LOF causes developmental delay, with homozygote NimA mutant larvae pupariating around one day later than heterozygotes (wild-type controls are missing and must be included). This delay would cause a 96h old larva to look like a 72h old larva, which are considerably thinner. As a consequence, the extent of proliferation and Dpn+ cell number would correspond to that of younger larvae, invalidating their claim. To solve this problem, the authors could measure brain size, cell proliferation and Dpn+ cell number in the mutants 24h later than in wild-type controls (if 24h is the period by which mutants are delayed, otherwise adjust accordingly) - or explain whether in fact this is what they did.

To measure the incidence of cell death, Dcp1 must be analyzed before the onset of an overt phenotype of decreased CNS size. That is, if the CNS is already smaller, then cell death already happened and will no longer be detectable. Apoptosis and the clearance of dead cells are both fast processes, thus anti-Dcp1+ cells are detectable within a narrow time window. This means that to quantify Dcp1+ cells, accurate staging needs to be used in order to obtain reliable data. Additionally, it can be challenging to identify the timing of cell death. The authors must repeat these experiments before the CNSs become overtly smaller, or rescue cell death e.g. with p35, as an alternative approach, or remove the claim.

To improve clarity, we implemented the information on the larval staging for CNS phenotypic analysis in the Methods sections (**line 414-416**) and more details are provided below. Notably, the developmental delay observed in *NimA* mutants or knockdown animals does not affect all individuals, some larvae still pupariate on the same day as controls (**Figure 4G**). Moreover, to ensure that the CNS phenotypes were not due to differences in developmental timing, dissections were performed specifically on wandering L3 larvae, a phase that starts at ~120 h in controls and lasts only few hours. Larvae were collected from the vial walls rather than from the medium where younger (e.g., 96 h) larvae remain feeding. As an additional confirmation of appropriate staging, no differences in larval size were observed between *NimA*-deficient and control animals. When necessary, the collection of *NimA* mutant or knockdown larvae was delayed by a few hours to obtain sufficient numbers of properly staged animals. Therefore, the larvae compared in **Figure 4A-F** are at the same developmental stage.

Concerning the incidence of cell death, we agree with the reviewer that the timing of cell death events is critical and assessing it only at the end point does not provide a comprehensive analysis. In the revised manuscript, we now state that the absence of detectable cell death at wL3 does not exclude the possibility that cell death may occur at earlier developmental stages (**line 186**).

Minor points:

6) Figure 5M should be mentioned within the results section, rather than only in the Discussion.

As suggested by the reviewer, **Figure 6** (Figure 5M in the previous version) has been mentioned in the result section (**line 100**).

7) Line 218 page 8, Discussion: "Our transcriptomic approach identifies a panel of 129 genes constitutive of a robust molecular signature of glia". As explained above, not sufficient evidence and information have been provided in this manuscript to support this statement. See above, point 1.

The strategy used to identify glia-specific genes has been clarified in the Results and Method sections (see response to comment 1).

8) Line 243-244, page 9 Discussion: "The high conservation of the NimA extracellular domain also suggests specific conserved ligands activating the receptor." However, Figures 3 and 5 suggest a function for NimA in homophilic cell adhesion - which would not necessitate any other ligand.

We thank the reviewer for raising this issue, which we agree upon. At this point, we cannot exclude the action of other ligands or interactors, as this seems supported by our observations. The EMI domain is crucial for promoting cell clumping in S2 cells (**Figure 5H-J**), yet it is not required for the formation of NimA multimers (**Figure 5N**). We have revised the discussion (**lines 314-319**) to improve clarity.

9) Line 250-252, page 9 Discussion: "page 9 Discussion: "Since several conserved tyrosine residues are also predicted to be phosphorylated in the intracellular domain of NimA (Figure 5M), it is possible that its multimerization leads to tyrosine phosphorylation to activate the downstream cascade responsible for cell aggregation". Wouldn't cell-to-cell adhesion (i.e. at the PM) suffice?

We agree with the reviewer that cell-cell adhesion could be sufficient to explain the observed phenotype. However, a role for the intracellular domain, similar to that described for other Nim family members, such as Drpr, is possible as well. In support of this, we obtained recent data highlighting the importance of the NimA intracellular domain: overexpression of NimA lacking this domain (NimA Δ intra) does not induce S2 cell aggregation (two independent trials, each analyzing >8000 cells; chi-square test comparing the total of big and small clumps in control vs. NimA Δ intra, P-value = 0.88).

Unfortunately, technical issues in isolating the NimA Δ intra protein for Co-IP and Western blot assays prevented further investigation of its molecular function (e.g., NimA multimerization or intracellular signaling). For this reason, we did not include these data in the revised manuscript. Should the referee feel that they are to be included, we can provide the figures.

10) Figure 3 is very nice. Figure 3B shows NimA to be expressed in surface glia/BBB surrounding the optic lobes but not the VNC. Why not? Similarly, the breakdown of BBB affects optic lobes in Figure 3E-F. Does it also occur in the VNC? Regarding Figure 3I, It would be nice to see the same magnification (ie whole CNS) for the NimA-KOGal4 mutant too. Does the phenotype in K happen also in the VNC? If not, could they speculate why?

Concerning NimA expression and function in the BBB of the VNC, please refer to the response to comment 5 from Referee #1.

As the reviewer will notice from the images below, the NrxFVGF signal is not very informative at low magnification, particularly in the brain lobes. For this reason, we relied on IMARIS-based 3D reconstruction to analyze septate junction organization and quantify multiple structural parameters. Because the low-magnification NrxFVGF images do not provide additional interpretable information, we did not include a *NimA KO-Gal4* image in **Figure 3I-K**.

11) Figure 3C-D are not mentioned within the text.

We apologize for this oversight, **Figure 3C-D** is now mentioned in **line 164** in the revised manuscript.

Reply to referee 1 in blue

Referee 1:

Sakr et al., revision

I appreciate the effort the authors took to address the reviewer comments. However, some of the new data images are confusing to this reviewer.

In Figure 1G,G, the activity of the NimA Gal4 knockin strain is shown. Here I recognize expression in the BBB and ensheathing glia and peripheral nerves (which oddly appear to be carefully removed from the image stack shown). In contrast in Figure EV2D-F expression of a different reporter (RFP) is indeed not found in the PNS. Why is here a difference? And how is expression of the RFP marker in the remaining brain?

As mentioned in the fig legend, the difference is coming from the confocal sections presented in the different panels. Figure 1G represents a single confocal section focusing on the median of the nervous system to cut through the ventral nerve cord and the optic lobes while Figure EV2D-F are substacks focused on the peripheral nerves taking roots in the dorsal part of the VNC (see schematic below).

Here below we produce comparable figures with matching focus showing that both eGFP and mCD8-RFP reporters display the same expression patterns. We believe that what is interpreted by the reviewer as NimA expression in peripheral nerves is signals from tract ensheathing glia (TEG) that are sending extensions to the periphery.

Substack Confocal section
on the nerve roots shown in Figure 1G

Similarly, when looking at Figure 2K almost no expression of NimA is seen in the SPG using scRNA seq data of the adult brain. In contrast in the rebuttal letter the values look different. The Figure in the rebuttal letter is also used to demonstrate very low expression of NimA in adult SPGs. If anything can be taken from this Figure it would be that in the adult only few cells express NimA and when looking at the original scRNA seq data - the NimA positive cells do not express the SPG marker *mdr65*. So at least in adults - which is not the focus of this study NimA is likely not expressed by SPGs.

We agree with the reviewer that NimA expression in the SPG is not clear from the scRNAseq data nor from the IF data shown in the rebuttal. In the manuscript, we use these data to show expression in EG and CG. This reinforces the authors' choice not to pursue adult BBB analysis.

Concerning the difference in between the rebuttal and Figure 2K, the only difference is that Fig 2K include a colour gradient for the expression levels, while the rebuttal figure only shows the cells in which NimA is detected (see below).

Figure 2

K scRNAseq adult CNS (Davie *et al.*, 2018)

Rebuttal letter - figure comment 3

To address this for the larval stage the authors provide technically challenging in situ hybridizations. From the description of the new Figure 3C,D I was not sure whether in situ hybridization has been performed with both *repo* and *NimA* in situ probes? In any case I find the data used to claim expression of *NimA* in larval SPGs not convincing. I would describe it as expression (white dots) outside the *moody-Gal4* positive cell (green), so most likely a cortex glia. Figure EV2G-H does not show a clearly recognizable brain. The GFP signal is missing and the argument for assigning the different glial cell type identities is not clear.

The new figure 3CD does show triple fluorescent in situ hybridisation using probes targeting *repo*, *GFP* and *NimA* as mention in the figure legend as well as in the method section. The position of the cell expressing *NimA* and *repo* on the outer periphery of the brain indicate that it can only be SPG or PG . The SPG identity is confirmed by the GFP signal driven by *moody-Gal4*, (which completely excludes the PG cell) and is consistent with the flat morphology of the nucleus, typical of SPG cells. We cannot exclude perdurance of endogenous GFP signal which would explain also why *Repo* and *NimA* transcript signals do not perfectly colocalize with the GFP signal. Also note that the green signals show GFP mRNA, not mCD8-GFP protein and thus do not show the complete boundary of the cell.

Figure 3

A slight disruption of septate junctions is described. However, the analysis is made in homozygous mutant NimA-KO flies, and as indicated earlier must be tested in trans to a deficiency or other null allele. In addition, the phenotype could be rescued by re-expressing NimA. The tools should be available.

We acknowledge the absence of this experiment, however, the data from NimA-KO flies and from glia-specific NimA silencing with two independent RNAi lines show that BBB function is compromised, strongly suggesting underlying structural defects.

Comments to the rebuttal letter:

The characterized NimA mutant alleles (which, like ours, display the "jaunty" curved-wing phenotype) carry only point mutations or very small deletions. As a result, they are likely to produce truncated proteins possibly retaining partial function. In contrast, our CRISPR/Cas9 NimA mutant is a true null, as the entire coding sequence has been removed. Because we do not know which domains of NimA are required for BBB integrity or CNS development, trans-heterozygous combinations using possibly hypomorphic alleles could produce false-negative results simply because the critical domains remain intact in those backgrounds. This consideration is further supported by our observation that silencing NimA in specific glial subtypes (ensheathing, cortex, or subperineurial glia) does not reproduce the jaunty wing phenotype, indicating that the jaunty phenotype is not necessarily linked to NimA's role in BBB formation or CNS development.

I do not agree to this response. The knockin allele must be analyzed in trans to a known null allele. When the authors have arguments that the existing NimA mutants are hypomorphs they should use a chromosomal deficiency. The notion that glial knockdown of NimA does not reproduce the wing phenotype does not support anything except that either the RNAi is ineffective or wings are unlinked to CNS glia.

We agree with the reviewer on the need of independent tools to validate phenotypes observed with the newly generated null NimA mutant. This is why, for the revision, we performed an extensive conditional knock down *in vivo* analysis and we even used drivers that target distinct glial subtypes. Using those drivers in combination with (two distinct) RNAi lines confirms our initial data. Furthermore, we also reveal subtype-specific phenotypes. We hope the reviewer will agree that this approach validates the NimA mutant generated in our study.

To rule out the possibility that the BBB phenotypes observed in the CRISPR/Cas9 NimA mutant arise from off-target effects, while simultaneously determining the glial subtype in which NimA is required, we expressed a NimA RNAi line in distinct glial populations using established drivers for subperineurial glia (Mdr65-GAL4, Mayer et al., 2009; Pfeiffer et al., 2010), perineurial glia (shn-GAL4, Pfeiffer et al., 2010; Kozlov et al., 2020), and cortex glia (VGlut2-GAL4, Pfeiffer et al., 2010; Kremer et al., 2017) and we analyzed the BBB phenotypes. Only the knockdown in subperineurial glia recapitulates the increased BBB permeability observed in the NimA mutant. (Figure 4H, data already present in the manuscript), similar to that observed in the mutant.

Unfortunately this is not shown in the Figure. The Figure 4H shows the percentage of pupae formed and not the increased BBB permeability.

We apologize for our mistake, the BBB permeability data are presented in figure 3H (and below), not 4H:

Dear Pierre,

Thank you for the submission of your revised manuscript. We have now received the enclosed reports from the referees, as well as cross-comments. Both referees still have comments that will need to be addressed. As you will see, referee 3 does not agree with all comments raised by referee 1, so please address all remaining comments along the lines suggested by referee 3. Please also submit a point-by-point response to all last comments.

A few editorial requests will also need to be addressed before we can proceed with the official acceptance of your manuscript:

- Please add up to 5 keywords to your ms file.
- The conflict of interest subheading needs to be corrected to "Disclosure and Competing Interests Statement"
- The term "unpublished observations" on page 4 is not allowed per journal policy. Please re-phrase or delete.
- The FUNDING INFO is not congruent; the following acronyms and funders are missing in our online ms submission system: UDS, CEFIPRA, Hôpital de Strasbourg, IdEx Unistra (ANR-10-IDEX-0002), SFRI-STRAT'US project (ANR 20-SFRI-0012), EUR IMCBio (ANR-17-EURE-0023) under the framework of the French Investments for the Future Program.
- For the FIGURE CALLOUTS 'Supplementary' may not be used, please correct.
- Table EV1 and Table EV2 are Datasets and need to be updated as such and updated to Dataset EV1 and Dataset EV2 in all places (source file names, titles in the system, legends, callouts in the ms text); the legends need to be removed from the ms and each legend should be provided in its corresponding Excel file in a new tab/sheet.
- In the Data Availability Section only data that was newly generated in this study and deposited in public databases should be listed, including with a direct html link to these data. Please remove all other information in this section.
- Materials and methods should be Methods.
- Please note that the data citations need to be tagged with the label "DATASET" in the reference list, please correct.

* Figure Legends - Comments *

- Please note that the exact p values are not provided in the legends of figures 3H, 5F, EV4 E, please provide exact values as reasonable.
- Please indicate the statistical test used for data analysis in the legend of figure 1B
- Please note that the box plots need to be defined in terms of minima, maxima, centre, bounds of box and whiskers, and percentile in the legends of figures 2I, J; 3G, H; 4C, D, E, F; M-O; EV3 J
- Please note that the error bars are not defined in the legends of figures 3L, M; 4K, L; 5F, J; EV1 D, EV4 G

I would like to suggest a few minor changes to the abstract. Please let me know whether you agree with this:

Glial cells are crucial for nervous system development and function by clearing debris, protecting neurons and ensuring neuronal survival. In *Drosophila*, glia form the blood-brain barrier, which regulates neural stem cell proliferation and shields the nervous system while maintaining communication with the rest of the organism. To uncover glial-specific roles, we here compare their transcriptome with that of neurons and macrophages. Our study identifies NimA, an uncharacterized member of the Nimrod family, as a glial-specific protein expressed during development. Unlike other family members (i.e. NimC1, Draper and NimC4/Simu) NimA is not involved in phagocytosis. Instead, NimA regulates cell-cell adhesion, crucial for maintaining the tight septate junctions of the larval BBB. Loss of NimA in BBB-forming glia compromises barrier integrity. Moreover, loss of NimA in those glia, or in glia that serve as neural stem cell niche, delays development, reduces brain size, impairs proliferation and reduces the neural stem cell pool. The identification of the glial-specific molecular landscape, including novel molecular players such as NimA, is key for understanding the contribution of glia to the nervous system.

EMBO press papers are accompanied online by A) a short (1-2 sentences) summary of the findings and their significance, B) 2-3 bullet points highlighting key results and C) a synopsis image that is exactly 550 pixels wide and 200-600 pixels high (the height is variable). The synopsis image should provide a sketch of the major findings, like a graphical abstract. Please note that text needs to be readable at the final size. Please send us this information along with the final manuscript.

Referee #1:

Sakr et al., revision

I appreciate the effort the authors took to address the reviewer comments. However, some of the new data images are confusing to this reviewer. In Figure 1G,G, the activity of the NimA Gal4 knockin strain is shown. Here I recognize expression in the BBB and ensheathing glia and peripheral nerves (which oddly appear to be carefully removed from the image stack shown). In contrast in Figure EV2D-F expression of a different reporter (RFP) is indeed not found in the PNS. Why is there a difference? And how is expression of the RFP marker in the remaining brain?

Similarly, when looking at Figure 2K almost no expression of NimA is seen in the SPG using scRNA seq data of the adult brain. In contrast in the rebuttal letter the values look different. The Figure in the rebuttal letter is also used to demonstrate very low expression of NimA in adult SPGs. If anything can be taken from this Figure it would be that in the adult only few cells express NimA and when looking at the original scRNA seq data - the NimA positive cells do not express the SPG marker *mdr65*. So at least in adults - which is not the focus of this study NimA is likely not expressed by SPGs.

To address this for the larval stage the authors provide technically challenging in situ hybridizations. From the description of the new Figure 3C,D I was not sure whether in situ hybridization has been performed with both *repo* and NimA in situ probes? In any case I find the data used to claim expression of NimA in larval SPGs not convincing. I would describe it as expression (white dots) outside the *moody-Gal4* positive cell (green), so most likely a cortex glia. Figure EV2G-H does not show a clearly recognizable brain. The GFP signal is missing and the argument for assigning the different glial cell type identities is not clear.

A slight disruption of septate junctions is described. However, the analysis is made in homozygous mutant NimA-KO flies, and as indicated earlier must be tested in trans to a deficiency or other null allele. In addition, the phenotype could be rescued by re-expressing NimA. The tools should be available.

Comments to the rebuttal letter:

The characterized NimA mutant alleles (which, like ours, display the "jaunty" curved-wing phenotype) carry only point mutations or very small deletions. As a result, they are likely to produce truncated proteins possibly retaining partial function. In contrast, our CRISPR/Cas9 NimA mutant is a true null, as the entire coding sequence has been removed. Because we do not know which domains of NimA are required for BBB integrity or CNS development, trans-heterozygous combinations using possibly hypomorphic alleles could produce false-negative results simply because the critical domains remain intact in those backgrounds. This consideration is further supported by our observation that silencing NimA in specific glial subtypes (ensheathing, cortex, or subperineurial glia) does not reproduce the jaunty wing phenotype, indicating that the jaunty phenotype is not necessarily linked to NimA's role in BBB formation or CNS development.

I do not agree to this response. The knockin allele must be analyzed in trans to a known null allele. When the authors have arguments that the existing NimA mutants are hypomorphs they should use a chromosomal deficiency. The notion that glial knockdown of NimA does not reproduce the wing phenotype does not support anything except that either the RNAi is ineffective or wings are unlinked to CNS glia.

To rule out the possibility that the BBB phenotypes observed in the CRISPR/Cas9 NimA mutant arise from off-target effects, while simultaneously determining the glial subtype in which NimA is required, we expressed a NimA RNAi line in distinct glial populations using established drivers for subperineurial glia (*Mdr65-GAL4*, Mayer et al., 2009; Pfeiffer et al., 2010), perineurial glia (*shn-GAL4*, Pfeiffer et al., 2010; Kozlov et al., 2020), and cortex glia (*VGlut2-GAL4*, Pfeiffer et al., 2010; Kremer et al., 2017) and we analyzed the BBB phenotypes. Only the knockdown in subperineurial glia recapitulates the increased BBB permeability observed in the NimA mutant. (Figure 4H, data already present in the manuscript), similar to that observed in the mutant. Unfortunately this is not shown in the Figure. The Figure 4H shows the percentage of pupae formed and not the increased BBB permeability.

Referee #3:

The authors have greatly improved the manuscript, have revised many figures, edited the text extensively and have addressed satisfactorily most of the points I raised. The manuscript is greatly improved and will be an interesting contribution to glial cell

biology.

With regards my previous raised criticisms:

Point 1: information has been included.

Point 2: very nice new data that demonstrate that in vivo NimA is not required for the clearance of crz neurons.

Point 3: has been resolved with excellent new quantifications of data, and including a suitable control.

Point 4 has been worked around with conditional knockdown in different glial cell types. This is acceptable, although it would improve by demonstrating that the RNAi lines did knock-down gene expression (eg with qRT-PCR).

Point 5: the staging is fine, thanks for the detailed explanation. However, the criticism on the claim on cell death has not been resolved.

Lines 183-186: "Accordingly, the NimA KO-Gal4 mutants show reduced CNS size at wL3 (Figure 4A-C). This phenotype is not associated with increased cell death at this stage, as the number of apoptotic cells detected by cleaved Dcp-1 immunolabeling (Song, McCall et al., 1997) is comparable to that observed in controls (Figure 4F)." This statement is still flawed. If the CNS is already of smaller size, then in order to test if this could be due to an increase in apoptosis, they needed to test apoptosis earlier on, before there was an overt reduction in CNS size. If loss of Dpn+ cells were to cause the reduction in CNS size, then Dcp1 would not reveal an increase in cell death, because Dpn+ cells would be already missing. Thus, to support their claim that NimA mutants do not affect apoptosis they need to look at an earlier time point, before there is overt cell loss (eg 24h earlier). Since they did not do this, then the evidence does not support the claim and the claim that NimA mutants do not affect apoptosis is unfounded and ought to be removed.

The authors also state " NimA KO-Gal4 mutants display a reduced neural stem cell pool and decreased cell proliferation which likely contribute to the smaller CNS size ". The reduction in Dpn+ cells in NimA mutants could not be due to a reduction in proliferation, as Dpn+ are neural stem cells that undergo self-renewing divisions (ie mother cell number does not increase). Instead, Dpn+ cell reduction could be due to cell fate determination deficits or loss of some Dpn+ cells. If there were fewer Dpn+ cells because cell fate determination is impaired, this would lead to reduced proliferation, but it would not mean that NimA regulates proliferation. Alternatively, proliferation deficits in NimA mutants would affect the generation of progeny cells. As it stands, the authors have not resolved why there are fewer Dpn+ cells in the mutants, nor which proliferating cells are affected by the mutants. They should add a comment on this point.

In Figure 4, they must show control and NimA KO-Gal4 CNSs stained with anti-DCp1, as they do for the other markers.

Altogether, the revision has improved the manuscript. The authors should tone down the above claims to improve the rigour of the paper.

Cross-comments from referee 3 on referee 1's comments:

In brief, the most important points are:

- The Reviewer is correct that the authors did not use a trans-heterozygous combination, which would have been best. In my view, obtaining phenotypes by knocking down in specific glial classes was an ok way to get around it (everyone does it...) - but they need to show the RNAi line works specifically for NimA.
- The Reviewer is wrong in the evaluation of the in situ data and the expression in Figure 3. They did not understand the experiment and what the figure shows.

More specific comments follow here:

Referee #1

Sakr et al., revision

I appreciate the effort the authors took to address the reviewer comments. However, some of the new data images are confusing to this reviewer. In Figure 1G,G, the activity of the NimA Gal4 knockin strain is shown. Here I recognize expression in the BBB and ensheathing glia and peripheral nerves (which oddly appear to be carefully removed from the image stack shown). In contrast in Figure EV2D-F expression of a different reporter (RFP) is indeed not found in the PNS. Why is here a difference? And how is expression of the RFP marker in the remaining brain?

Ref 3: It is present in the nerves, see EV2F'. It is fainter than in the CNS, but there is signal in nerves. In Figure 1G,G' there are no nerves, simply because this happens sometimes with the dissection, sometimes they ear off and sometimes they don't. As they show signal in nerves in EV2F' and the work does not focus more deeply on the nerves anyway, this point does not seem very relevant.

Similarly, when looking at Figure 2K almost no expression of NimA is seen in the SPG using scRNA seq data of the adult brain. In contrast in the rebuttal letter the values look different. The Figure in the rebuttal letter is also used to demonstrate very low expression of NimA in adult SPGs. If anything can be taken from this Figure it would be that in the adult only few cells express NimA and when looking at the original scRNA seq data - the NimA positive cells do not express the SPG marker *mdr65*. So at least in adults - which is not the focus of this study NimA is likely not expressed by SPGs.

Ref 3: Figure 2K,K' shows expression throughout the glial classes, including the cluster of PG and SPG. The Reviewer is correct that the figure provided in the rebuttal letter is different from Figure 2K,K', and includes more expressing cells. Maybe the authors had different settings on in Scope or they used a different database? It would be best if the authors could double check and confirm which is the more accurate expression and update the paper.

Having said that, the authors argue in the rebuttal letter that NimA functions primarily in development, and is only homeostatic in the adult. This seems ok, as genes frequently have different functions in development and in the adult. The authors should check that this point has been made clear within the text.

To address this for the larval stage the authors provide technically challenging in situ hybridizations. From the description of the new Figure 3C,D I was not sure whether in situ hybridization has been performed with both *repo* and NimA in situ probes? In any case I find the data used to claim expression of NimA in larval SPGs not convincing. I would describe it as expression (white dots) outside the *moody-Gal4* positive cell (green), so most likely a cortex glia.

Ref 3: the methods section clearly explains that these are triple in situs, ie detecting mRNA for the three genes. Reviewer 1 is not right in their interpretation that the signal is in cortex glia. The location of the mRNA signal for all the three probes is distal to the nuclei labelled with DAPI. There are no further cells between DAPI and the mRNA. Thus, the mRNA can only be in the outermost layer of glia (ie BBB). The sub-cellular distribution of the mRNAs is not identical, but there is no evidence that it ought to be.

Figure EV2G-H does not show a clearly recognizable brain. The GFP signal is missing and the argument for assigning the different glial cell type identities is not clear.

Ref 3: Reviewer 1 is correct that GFP channel is missing. And that the images are not great.

A slight disruption of septate junctions is described. However, the analysis is made in homozygous mutant NimA-KO flies, and as indicated earlier must be tested in trans to a deficiency or other null allele. In addition, the phenotype could be rescued by re-expressing NimA. The tools should be available.

Ref 3: the authors have dealt with this.

Comments to the rebuttal letter:

The characterized NimA mutant alleles (which, like ours, display the "jaunty" curved-wing phenotype) carry only point mutations or very small deletions. As a result, they are likely to produce truncated proteins possibly retaining partial function. In contrast, our CRISPR/Cas9 NimA mutant is a true null, as the entire coding sequence has been removed. Because we do not know which domains of NimA are required for BBB integrity or CNS development, trans-heterozygous combinations using possibly hypomorphic alleles could produce false-negative results simply because the critical domains remain intact in those backgrounds. This consideration is further supported by our observation that silencing NimA in specific glial subtypes (ensheathing, cortex, or subperineurial glia) does not reproduce the jaunty wing phenotype, indicating that the jaunty phenotype is not necessarily linked to NimA's role in BBB formation or CNS development.

I do not agree to this response. The knockin allele must be analyzed in trans to a known null allele. When the authors have arguments that the existing NimA mutants are hypomorphs they should use a chromosomal deficiency. The notion that glial knockdown of NimA does not reproduce the wing phenotype does not support anything except that either the RNAi is ineffective or wings are unlinked to CNS glia.

Ref 3: I agree with Reviewer 1 that using only homozygous alleles is not good. It is correct that the authors could have use transheterozygotes over a deficiency. However, in my view using the RNAis was an acceptable solution, done by many labs. It would best if they could validate that the RNAi works (eg with qRT-PCR).

To rule out the possibility that the BBB phenotypes observed in the CRISPR/Cas9 NimA mutant arise from off-target effects,

while simultaneously determining the glial subtype in which NimA is required, we expressed a NimA RNAi line in distinct glial populations using established drivers for subperineurial glia (Mdr65-GAL4, Mayer et al., 2009; Pfeiffer et al., 2010), perineurial glia (shn-GAL4, Pfeiffer et al., 2010; Kozlov et al., 2020), and cortex glia (VGlut2-GAL4, Pfeiffer et al., 2010; Kremer et al., 2017) and we analyzed the BBB phenotypes. Only the knockdown in subperineurial glia recapitulates the increased BBB permeability observed in the NimA mutant. (Figure 4H, data already present in the manuscript), similar to that observed in the mutant. Unfortunately this is not shown in the Figure. The Figure 4H shows the percentage of pupae formed and not the increased BBB permeability.

Ref 3: I agree that Figure 4H does not show increased BBB permeability. The authors should either carry out a BBB permeability test, or change their claim to make it more rigorous and centered on what the data show (ie Only knock-down in subperineurial glia affects pupariation).

Latest comments from referee 1 on authors reply:

Frankly, I am a bit surprised by the letter. A few things are trivial such as: Figure 1G versus Figure EV2D-F. This is now better explained, but I am not sure how they put this in their manuscript. The Figure 2K shows correctly the scRNA data from the Aerts lab. The rebuttal letter shows a different image. The statement that Figure 2K includes a color gradient for the expression levels, while the rebuttal figure only shows the cells in which NimA is detected is not quite correct. In the rebuttal the very few NimA expressing cells are enlarged enormously and the annotation is wrong (blue color does not mean subperineurial glia).

More seriously

Figure 3 does not show expression of NimA (white) in subperineurial glia (green). the two colors are in different cells. I really wonder how the authors can ignore this.

Lastly, analysis of a transheterozygous mutant. This is an extremely easy experiment. Deficiencies are available at Bloomington and any deficiency acts as a bona fide null allele! So this is the easiest experiment to do and I more and more wonder why the authors refuse to do it. It really should be done-

Ref 3: I disagree with these two comments - 4 paragraphs, all on the same point. Firstly, it does actually matter that NimA is expressed at the larval BBB, and this has been demonstrated well. The authors clearly show with in situ hybridisations that NimA is expressed in SG and PG in the larval CNS (Figure 3C-D'). Reviewer 1 did not understand these images. The colours are not in different cells: Reviewer 1 missed the position of the nuclei (DAPI). They missed that the labelled mRNAs are within the same cytoplasm. There is no reason or evidence to believe that all mRNAs need to be exactly colocalised, in fact, they are not (eg localised translation). Secondly, there are frequent discrepancies between scRNAseq and other visualisation data (in situ, antibody stainings), and between mRNA and protein. The Scope scRNAseq data are a snapshot of the adult brain. The scRNAseq results do not cancel the larval in situ. The Scope data themselves will have error, and only contain a fraction of the total number of cells in the brain.

Reply to referee 1 in blue

Referee 1:

Sakr et al., revision

I appreciate the effort the authors took to address the reviewer comments. However, some of the new data images are confusing to this reviewer.

In Figure 1G,G, the activity of the NimA Gal4 knockin strain is shown. Here I recognize expression in the BBB and ensheathing glia and peripheral nerves (which oddly appear to be carefully removed from the image stack shown). In contrast in Figure EV2D-F expression of a different reporter (RFP) is indeed not found in the PNS. Why is here a difference? And how is expression of the RFP marker in the remaining brain?

As mentioned in the fig legend, the difference is coming from the confocal sections presented in the different panels. Figure 1G represents a single confocal section focusing on the median of the nervous system to cut through the ventral nerve cord and the optic lobes while Figure EV2D-F are substacks focused on the peripheral nerves taking roots in the dorsal part of the VNC (see schematic below).

Here below we produce comparable figures with matching focus showing that both eGFP and mCD8-RFP reporters display the same expression patterns. We believe that what is interpreted by the reviewer as NimA expression in peripheral nerves is signals from tract ensheathing glia (TEG) that are sending extensions to the periphery.

Similarly, when looking at Figure 2K almost no expression of NimA is seen in the SPG using scRNA seq data of the adult brain. In contrast in the rebuttal letter the values look different. The Figure in the rebuttal letter is also used to demonstrate very low expression of NimA in adult SPGs. If anything can be taken from this Figure it would be that in the adult only few cells express NimA and when looking at the original scRNA seq data - the NimA positive cells do not express the SPG marker *mdr65*. So at least in adults - which is not the focus of this study NimA is likely not expressed by SPGs.

We agree with the reviewer that NimA expression in the SPG is not clear from the scRNAseq data nor from the IF data shown in the rebuttal. In the manuscript, we use these data to show expression in EG and CG. This reinforces the authors' choice not to pursue adult BBB analysis.

Concerning the difference in between the rebuttal and Figure 2K, the only difference is that Fig 2K include a colour gradient for the expression levels, while the rebuttal figure only shows the cells in which NimA is detected (see below).

Figure 2

K scRNAseq adult CNS (Davie *et al.*, 2018)

Rebuttal letter - figure comment 3

To address this for the larval stage the authors provide technically challenging in situ hybridizations. From the description of the new Figure 3C,D I was not sure whether in situ hybridization has been performed with both *repo* and *NimA* in situ probes? In any case I find the data used to claim expression of *NimA* in larval SPGs not convincing. I would describe it as expression (white dots) outside the *moody-Gal4* positive cell (green), so most likely a cortex glia. Figure EV2G-H does not show a clearly recognizable brain. The GFP signal is missing and the argument for assigning the different glial cell type identities is not clear.

The new figure 3CD does show triple fluorescent in situ hybridisation using probes targeting *repo*, *GFP* and *NimA* as mention in the figure legend as well as in the method section. The position of the cell expressing *NimA* and *repo* on the outer periphery of the brain indicate that it can only be SPG or PG . The SPG identity is confirmed by the GFP signal driven by *moody-Gal4*, (which completely excludes the PG cell) and is consistent with the flat morphology of the nucleus, typical of SPG cells. We cannot exclude perdurance of endogenous GFP signal which would explain also why *Repo* and *NimA* transcript signals do not perfectly colocalize with the GFP signal. Also note that the green signals show GFP mRNA, not mCD8-GFP protein and thus do not show the complete boundary of the cell.

Figure 3

A slight disruption of septate junctions is described. However, the analysis is made in homozygous mutant NimA-KO flies, and as indicated earlier must be tested in trans to a deficiency or other null allele. In addition, the phenotype could be rescued by re-expressing NimA. The tools should be available.

We acknowledge the absence of this experiment, however, the data from NimA-KO flies and from glia-specific NimA silencing with two independent RNAi lines show that BBB function is compromised, strongly suggesting underlying structural defects.

Comments to the rebuttal letter:

The characterized NimA mutant alleles (which, like ours, display the "jaunty" curved-wing phenotype) carry only point mutations or very small deletions. As a result, they are likely to produce truncated proteins possibly retaining partial function. In contrast, our CRISPR/Cas9 NimA mutant is a true null, as the entire coding sequence has been removed. Because we do not know which domains of NimA are required for BBB integrity or CNS development, trans-heterozygous combinations using possibly hypomorphic alleles could produce false-negative results simply because the critical domains remain intact in those backgrounds. This consideration is further supported by our observation that silencing NimA in specific glial subtypes (ensheathing, cortex, or subperineurial glia) does not reproduce the jaunty wing phenotype, indicating that the jaunty phenotype is not necessarily linked to NimA's role in BBB formation or CNS development.

I do not agree to this response. The knockin allele must be analyzed in trans to a known null allele. When the authors have arguments that the existing NimA mutants are hypomorphs they should use a chromosomal deficiency. The notion that glial knockdown of NimA does not reproduce the wing phenotype does not support anything except that either the RNAi is ineffective or wings are unlinked to CNS glia.

We agree with the reviewer on the need of independent tools to validate phenotypes observed with the newly generated null NimA mutant. This is why, for the revision, we performed an extensive conditional knock down *in vivo* analysis and we even used drivers that target distinct glial subtypes. Using those drivers in combination with (two distinct) RNAi lines confirms our initial data. Furthermore, we also reveal subtype-specific phenotypes. We hope the reviewer will agree that this approach validates the NimA mutant generated in our study.

To rule out the possibility that the BBB phenotypes observed in the CRISPR/Cas9 NimA mutant arise from off-target effects, while simultaneously determining the glial subtype in which NimA is required, we expressed a NimA RNAi line in distinct glial populations using established drivers for subperineurial glia (Mdr65-GAL4, Mayer et al., 2009; Pfeiffer et al., 2010), perineurial glia (shn-GAL4, Pfeiffer et al., 2010; Kozlov et al., 2020), and cortex glia (VGlut2-GAL4, Pfeiffer et al., 2010; Kremer et al., 2017) and we analyzed the BBB phenotypes. Only the knockdown in subperineurial glia recapitulates the increased BBB permeability observed in the NimA mutant. (Figure 4H, data already present in the manuscript), similar to that observed in the mutant.

Unfortunately this is not shown in the Figure. The Figure 4H shows the percentage of pupae formed and not the increased BBB permeability.

We apologize for our mistake, the BBB permeability data are presented in figure 3H (and below), not 4H:

Dear Esther,

We sincerely thank you and Referee #3 for the thorough assessment of our manuscript, including the time and effort devoted to the cross-commenting process.

Below, we present a point-by-point reply to all last comments. Revisions introduced in the manuscript are highlighted in yellow.

Referee #3:

The authors have greatly improved the manuscript, have revised many figures, edited the text extensively and have addressed satisfactorily most of the points I raised. The manuscript is greatly improved and will be an interesting contribution to glial cell biology.

We appreciate the reviewer's favorable assessment and constructive feedback.

With regards my previous raised criticisms:

Point 1: information has been included.

Point 2: very nice new data that demonstrate that in vivo NimA is not required for the clearance of crz neurons.

Point 3: has been resolved with excellent new quantifications of data, and including a suitable control.

Point 4 has been worked around with conditional knockdown in different glial cell types. This is acceptable, although it would improve by demonstrating that the RNAi lines did knock-down gene expression (eg with qRT-PCR).

As suggested by the reviewer, the efficiency of the two NimA RNAi lines used in this study was assessed by qPCR. RNA was extracted from whole wL3 larvae following ubiquitous NimA knockdown using either of the two independent RNAi lines (*Ubi-Gal4/NimA RNAi 104204* and *Ubi-Gal4/NimA RNAi 105009*) and compared with controls (*Ubi-Gal4/+*). $n = 3$, statistics: bilateral student test for equal variance. In both cases, *NimA* expression levels are significantly reduced (revised **Figure EV4A** and shown below), validating this approach as suitable for investigating NimA function upon its loss in specific glial cell types.

Point 5: the staging is fine, thanks for the detailed explanation. However, the criticism on the claim on cell death has not been resolved.

Lines 183-186: "Accordingly, the NimA KO-Gal4 mutants show reduced CNS size at wL3 (Figure 4A-C). This phenotype is not associated with increased cell death at this stage, as the number of apoptotic cells detected by cleaved Dcp-1 immunolabeling (Song, McCall et al., 1997) is comparable to that observed in controls (Figure 4F)." This statement is still flawed. If the CNS is already of smaller size, then in order to test if this could be due to an increase in apoptosis, they needed to test apoptosis earlier on, before there was an overt reduction in CNS size. If loss of Dpn+ cells were to cause the reduction in CNS size, then Dcp1 would not reveal an increase in cell death, because Dpn+ cells would be already missing. Thus, to support their claim that NimA mutants do not affect apoptosis they need to look at an earlier time point, before there is overt cell loss (eg 24h earlier). Since they did not do this, then the evidence does not support the claim and the claim that NimA mutants do not affect apoptosis is unfounded and ought to be removed.

As suggested by the reviewer, the claim has been removed from the revised manuscript, along with the Dcp-1 quantification in *NimA-KO-Gal4* mutant CNS previously shown in Figure 4.

The authors also state " NimA KO-Gal4 mutants display a reduced neural stem cell pool and decreased cell proliferation which likely contribute to the smaller CNS size ". The reduction in Dpn+ cells in NimA mutants could not be due to a reduction in proliferation, as Dpn+ are neural stem cells that undergo self-renewing divisions (ie mother cell number does not increase). Instead, Dpn+ cell reduction could be due to cell fate determination deficits or loss of some Dpn+ cells. If there were fewer Dpn+ cells because cell fate determination is impaired, this would lead to reduced proliferation, but it would not mean that NimA regulates proliferation. Alternatively, proliferation deficits in NimA mutants would affect the generation of progeny cells. As it stands, the authors have not resolved why there are fewer Dpn+ cells in the mutants, nor which proliferating cells are affected by the mutants. They should add a comment on this point.

We agree with the reviewer comment, and we added the following sentence to the discuss section "The basis of neural stem cell loss remains unresolved and may arise from defects in cell fate determination or cell survival (rather than impaired self-renewal), potentially leading to reduced proliferation of progeny cells." (line 295-297)

In Figure 4, they must show control and NimA KO-Gal4 CNSs stained with anti-DCp1, as they do for the other markers.

To address point 5 from Referee #3, Dcp-1 quantification in *NimA-KO-Gal4* mutant CNS has been removed from Figure 4.

Altogether, the revision has improved the manuscript. The authors should tone down the above claims to improve the rigour of the paper.

Cross-comments from referee 3 on referee 1's comments:

In brief, the most important points are:

- The Reviewer is correct that the authors did not use a trans-heterozygous combination, which would have been best. In my view, obtaining phenotypes by knocking down in specific glial classes was an ok way to get around it (everyone does it...) - but they need to show the RNAi line works specifically for NimA.

Please, see response to point 4 of Referee #3.

- The Reviewer is wrong in the evaluation of the in situ data and the expression in Figure 3. They did not understand the experiment and what the figure shows.

More specific comments follow here:

Referee #1

Sakr et al., revision

I appreciate the effort the authors took to address the reviewer comments. However, some of the new data images are confusing to this reviewer. In Figure 1G,G, the activity of the NimA Gal4 knockin strain is shown. Here I recognize expression in the BBB and ensheathing glia and peripheral nerves (which oddly appear to be carefully removed from the image stack shown). In contrast in Figure EV2D-F expression of a different reporter (RFP) is indeed not found in the PNS. Why is here a difference? And how is expression of the RFP marker in the remaining brain?

Ref 3: It is present in the nerves, see EV2F'. It is fainter than in the CNS, but there is signal in nerves. In Figure 1G,G' there are no nerves, simply because this happens sometimes with the dissection, sometimes they ear off and sometimes they don't. As they show signal in nerves in EV2F' and the work does not focus more deeply on the nerves anyway, this point does not seem very relevant.

Similarly, when looking at Figure 2K almost no expression of NimA is seen in the SPG using scRNA seq data of the adult brain. In contrast in the rebuttal letter the values look different. The Figure in the rebuttal letter is also used to demonstrate very low expression of NimA in adult SPGs. If anything can be taken from this Figure it would be that in the adult only few cells express NimA and when looking at the original scRNA seq data - the NimA positive cells do not express the SPG marker *mdr65*. So at least in adults - which is not the focus of this study NimA is likely not expressed by SPGs.

Ref 3: Figure 2K,K' shows expression throughout the glial classes, including the cluster of PG and SPG. The Reviewer is correct that the figure provided in the rebuttal letter is different from Figure 2K,K', and includes more expressing cells. Maybe the authors had different settings on in Scope or they used a different database? It would be best if the authors could double check and confirm which is the more accurate expression and update the paper.

We confirm that both Figure 2K,K' and the chart shown in the rebuttal letter are derived from exactly the same dataset available in Scope (Davie et al. 2018). However, while Figure 2K,K' display a color gradient representing NimA expression levels, the rebuttal figure is a screenshot from the Scope website and only indicates cells in which NimA is detected (on/off signal). As the data presented in

Figure 2K,K' provide a more accurate representation than the rebuttal figure, no changes have been made in this regard in the revised manuscript.

Having said that, the authors argue in the rebuttal letter that NimA functions primarily in development, and is only homeostatic in the adult. This seems ok, as genes frequently have different functions in development and in the adult. The authors should check that this point has been made clear within the text.

We believe this point is clearly stated in the manuscript: “Finally, our analyses indicate that NimA loss does not introduce further developmental delay or lethality past the larval stage, implying that NimA’s role is largely confined to developmental stages, with compensatory pathways likely preserving adult homeostasis.” (line 308-310) and “In sum, this study identifies a distinctive glial molecular profile, uncovering novel glial markers and a previously undescribed developmental role for NimA, an uncharacterized member of the Nimrod protein family, in BBB integrity and CNS development” (line 328-330), hence no modifications have been made in this regard.

To address this for the larval stage the authors provide technically challenging in situ hybridizations. From the description of the new Figure 3C,D I was not sure whether in situ hybridization has been performed with both repo and NimA in situ probes? In any case I find the data used to claim expression of NimA in larval SPGs not convincing. I would describe it as expression (white dots) outside the moody-Gal4 positive cell (green), so most likely a cortex glia.

Ref 3: the methods section clearly explains that these are triple in situs, ie detecting mRNA for the three genes. Reviewer 1 is not right in their interpretation that the signal is in cortex glia. The location of the mRNA signal for all the three probes is distal to the nuclei labelled with DAPI. There are no further cells between DAPI and the mRNA. Thus, the mRNA can only be in the outermost layer of glia (ie BBB). The sub-cellular distribution of the mRNAs is not identical, but there is no evidence that it ought to be.

Figure EV2G-H does not show a clearly recognizable brain. The GFP signal is missing and the argument for assigning the different glial cell type identities is not clear.

Ref 3: Reviewer 1 is correct that GFP channel is missing. And that the images are not great.

To improve clarity, we included the GFP channel and added a full-stack CNS projection showing all channels (Figure EV2H-I'''), with a yellow square indicating the region magnified and shown as a single section in (I-I'''). All channels are shown in (I), while individual GFP, NimA, and Repo channels are shown in (I'-I'''), where glial cell types are assigned based on their localization (as specified in the revised figure legend). To improve image quality, we reprocessed the raw images *de novo* using Fiji software. Although we attempted to center the same region, slight differences from the previous version remain. For simplicity, we removed the single-section DAPI image, as the dense packing of brain nuclei did not allow clear discrimination of individual nuclei. Both the original and revised versions are shown below. We hope the reviewer can appreciate the improvements.

PREVIOUS VERSION

REVISED VERSION

A slight disruption of septate junctions is described. However, the analysis is made in homozygous mutant *NimA*-KO flies, and as indicated earlier must be tested in trans to a deficiency or other null allele. In addition, the phenotype could be rescued by re-expressing *NimA*. The tools should be available.

Ref 3: the authors have dealt with this.

Comments to the rebuttal letter:

The characterized *NimA* mutant alleles (which, like ours, display the "jaunty" curved-wing phenotype) carry only point mutations or very small deletions. As a result, they are likely to produce truncated proteins possibly retaining partial function. In contrast, our CRISPR/Cas9 *NimA* mutant is a true null, as the entire coding sequence has been removed. Because we do not know which domains of *NimA* are required for BBB integrity or CNS development, trans-heterozygous combinations using possibly hypomorphic alleles could produce false-negative results simply because the critical domains remain intact in those backgrounds. This consideration is further supported by our observation that silencing *NimA* in specific glial subtypes (ensheathing, cortex, or subperineurial glia) does not reproduce the jaunty wing phenotype, indicating that the jaunty phenotype is not necessarily linked to *NimA*'s role in BBB formation or CNS development.

I do not agree to this response. The knockin allele must be analyzed in trans to a known null allele. When the authors have arguments that the existing *NimA* mutants are hypomorphs they should use a chromosomal deficiency. The notion that glial knockdown of *NimA* does not reproduce the wing phenotype does not support anything except that either the RNAi is ineffective or wings are unlinked to CNS glia.

Ref 3: I agree with Reviewer 1 that using only homozygous alleles is not good. It is correct that the authors could have use transheterozygotes over a deficiency. However, in my view using the RNAis

was an acceptable solution, done by many labs. It would best if they could validate that the RNAi works (eg with qRT-PCR).

Please, see response to point 4 of Referee #3.

To rule out the possibility that the BBB phenotypes observed in the CRISPR/Cas9 NimA mutant arise from off-target effects, while simultaneously determining the glial subtype in which NimA is required, we expressed a NimA RNAi line in distinct glial populations using established drivers for subperineurial glia (Mdr65-GAL4, Mayer et al., 2009; Pfeiffer et al., 2010), perineurial glia (shn-GAL4, Pfeiffer et al., 2010; Kozlov et al., 2020), and cortex glia (VGlut2-GAL4, Pfeiffer et al., 2010; Kremer et al., 2017) and we analyzed the BBB phenotypes. Only the knockdown in subperineurial glia recapitulates the increased BBB permeability observed in the NimA mutant. (Figure 4H, data already present in the manuscript), similar to that observed in the mutant. Unfortunately this is not shown in the Figure. The Figure 4H shows the percentage of pupae formed and not the increased BBB permeability.

Ref 3: I agree that Figure 4H does not show increased BBB permeability. The authors should either carry out a BBB permeability test, or change their claim to make it more rigorous and centered on what the data show (ie Only knock-down in subperineurial glia affects pupariation).

We apologize for our mistake in citing the pertinent figure, the BBB permeability data are presented in Figure 3H (and below), not in Figure 4H.

Latest comments from referee 1 on authors reply:

Frankly, I am a bit surprised by the letter. A few things are trivial such as:

Figure 1G versus Figure EV2D-F. This is now better explained, but I am not sure how they put this in their manuscript.

The Figure 2K shows correctly the scRNA data from the Aerts lab. The rebuttal letter shows a different image. The statement that Figure 2K includes a color gradient for the expression levels, while the rebuttal figure only shows the cells in which NimA is detected is not quite correct. In the rebuttal the very few NimA expressing cells are enlarged enormously and the annotation is wrong (blue color does not mean subperineurial glia).

More seriously

Figure 3 does not show expression of NimA (white) in subperineurial glia (green).
the two colors are in different cells. I really wonder how the authors can ignore this.

Lastly, analysis of a transheterozygous mutant. This is an extremely easy experiment. Deficiencies are available at Bloomington and any deficiency acts as a bona fide null allele! So this is the easiest experiment to do and I more and more wonder why the authors refuse to do it. It really should be done-

Ref 3: I disagree with these two comments - 4 paragraphs, all on the same point. Firstly, it does actually matter that NimA is expressed at the larval BBB, and this has been demonstrated well. The authors clearly show with in situ hybridisations that NimA is expressed in SG and PG in the larval CNS (Figure 3C-D'). Reviewer 1 did not understand these images. The colours are not in different cells: Reviewer 1 missed the position of the nuclei (DAPI). They missed that the labelled mRNAs are within the same cytoplasms. There is no reason or evidence to believe that all mRNAs need to be exactly colocalised, in fact, they are not (eg localised translation). Secondly, there are frequent discrepancies between scRNAseq and other visualisation data (in situs, antibody stainings), and between mRNA and protein. The Scope scRNAseq data are a snapshot of the adult brain. The scRNAseq results do not cancel the larval in situs. The Scope data themselves will have error, and only contain a fraction of the total number of cells in the brain.

Dr. Pierre Cattenoz
Institute of Genetics and Molecular and Cell Biology
Functional Genomics and Cancer
1 rue L Fries
ILLKRICH 67404
France

Dear Pierre,

I am very pleased to accept your manuscript for publication in the next available issue of EMBO reports. Thank you for your contribution to our journal.

You may qualify for financial assistance for your publication charges - either via a Springer Nature fully open access agreement or an EMBO initiative. Check your eligibility: <https://link.springer.com/journal/44319/how-to-publish-with-us>

>>> Please note that it is EMBO Reports policy for the transcript of the editorial process (containing referee reports and your response letter) to be published as an online supplement to each paper. If you do NOT want this, you will need to inform the Editorial Office via email immediately. More information is available here: <https://link.springer.com/partners/embo-press/editorial-policies#Peer%20review>